# LaGEA: Language Guided Embodied Agents for Robotic Manipulation

**Abdul Monaf Chowdhury** [1]  **Akm Moshiur Rahman Mazumder** [2]  **Safaeid Hossain Arib** [1]  **Rabeya Akter** [1]

## Abstract

Robotic manipulation benefits from foundation models that describe goals, but today's agents still lack a principled way to learn from their own mistakes. We ask whether natural language can serve as feedback, an error-reasoning signal that helps embodied agents diagnose what went wrong and correct course. We introduce LaGEA (**La**nguage **G**uided **E**mbodied **A**gents), a framework that turns episodic, schema-constrained reflections from a vision language model (VLM) into temporally grounded guidance for reinforcement learning. LaGEA summarizes each attempt in concise language, localizes the decisive moments in the trajectory, aligns feedback with visual state in a shared representation, and converts goal progress and feedback agreement into bounded, step-wise shaping rewards whose influence is modulated by an adaptive, failure-aware coefficient. This design yields dense signals early when exploration needs direction and gracefully recedes as competence grows. On the Meta-World MT10 and Robotic Fetch embodied manipulation benchmark, LaGEA improves average success over the state-of-the-art (SOTA) methods by 9.0% on random goals, 5.3% on fixed goals, and 17% on fetch tasks, while converging faster. These results support our hypothesis: language, when structured and grounded in time, is an effective mechanism for teaching robots to self-reflect on mistakes and make better choices. Code: https://github.com/monaf-chowdhury/LaGEA

## 1. Introduction

Multimodal foundation models have reshaped sequential decision-making (Yang et al., 2023), from language-grounded affordance reasoning (Ahn et al., 2022) to vision–language–action transfer, robots now display compelling zero-shot behaviour and semantic competence (Driess et al., 2023; Kim et al., 2024; Brohan et al., 2024). Yet converting such priors into reliable learning signals still hinges on reward design, which remains a bottleneck across tasks and scenes. To reduce engineering overhead, a pragmatic trend is to treat VLMs as zero-shot reward models (Rocamonde et al., 2023), scoring progress from natural-language goals and visual observations(Baumli et al., 2023). Yet these scores usually summarize overall outcomes rather than provide step-wise credit, can fluctuate with viewpoint and context, and inherit biases and inconsistency (Wang et al., 2022; Li et al., 2024).

Densifying VLM-derived rewards into per-step signals helps but does not remove hallucination or noise-induced drift. Simply adding these signals can destabilize training or encourage reward hacking. Contrastive objectives like FuRL (Fu et al., 2024) reduce reward misalignment, but on long-horizon, sparse-reward tasks, early misalignment can compound, misdirecting exploration. This highlights the need for structured, temporally grounded guidance that reduces noise and helps the agent recognize and learn from its own failures.

Agents need to recognize what went wrong, when it happened, and why it matters for the next decision. General-purpose VLMs, while capable at instruction-following, are not calibrated for this role, as they can hallucinate or rationalize errors under small distribution shifts (Lin et al., 2021). Prior self-reflection paradigms (Shinn et al., 2023) show that textual self-critique can improve decision making, but these demonstrations largely live in text-only environments such as ALFWorld (Shridhar et al., 2020), where observation, action, and feedback share a symbolic interface. Learning from failure is a fundamental aspect of reasoning; therefore, we ask a critical question: *How can embodied policies derive reliable, temporally localized failure attributions directly from visual trajectories of the stochastic robotic environments where explorations are expensive?*

Learning from mistakes requires detecting failures and causal understanding. For this purpose, we present our framework **LaGEA**, which addresses this by using VLMs to generate episodic natural-language reflections on a robot's

---

[1]Department of Robotics & Mechatronics Engineering, University of Dhaka, Bangladesh [2]Center for Computational & Data Sciences, Independent University, Bangladesh. Correspondence to: Abdul Monaf Chowdhury <monafabdul15@gmail.com>.

*Proceedings of the 43rd International Conference on Machine Learning*, Seoul, South Korea. PMLR 306, 2026. Copyright 2026 by the author(s).

behavior, summarizing what was attempted, which constraints were violated, and providing actionable rationales. As smaller VLMs can hallucinate or drift in free-form text (Guan et al., 2024; Chen et al., 2024b), feedback is structured and aligned with goal and instruction texts, making LAGEA transferable across agents, viewpoints, and environments while maintaining stability.

With these structured reflections in hand, we turn feedback into a signal the agent can actually use at each step rather than as a single episode score. LᴀGEA maps the feedback into the agent's visual representation and attaches a local progress signal to each transition. We adopt potential-based reward shaping, adding only the change in this signal from successive states, which avoids over-rewarding static states (Wiewiora, 2003). The potential itself blends two agreements: how well the current state matches the instruction-defined goal, and how well the transition aligns with the VLM's diagnosis around the key frames, so progress is rewarded precisely where the diagnosis says it matters. To keep learning stable, we dynamically modulate its scale against the environment task reward and feed the overall reward to the critic of our online RL algorithm (Haarnoja et al., 2018).

We evaluate LᴀGEA on diverse robotic manipulation tasks and transform VLM critique into localized, action-grounded shaping, obtains faster convergence and higher success rates over strong off-policy baselines. Our core contributions are:

- We present LᴀGEA, an embodied VLM-RL framework that generates causal episodic feedback which are localized in time to turn failures into guidance and improve recovery after near misses.

- We demonstrate that LᴀGEA can convert episodic, natural language self-reflection into a dense reward shaping signal through feedback alignment and feedback-VLM delta reward potential that can solve complex, sparse reward robot manipulation tasks.

- We provide extensive experimental analysis of LᴀGEA on state-of-the-art (SOTA) robotic manipulation benchmarks and present insights into LᴀGEA's learning procedure via thorough ablation studies.

## 2. Related Work

**VLMs for RL.** Foundation models (Wiggins & Tejani, 2022) have proven broadly useful across downstream applications (Khandelwal et al., 2022; Chowdhury et al., 2025), motivating their incorporation into reinforcement learning pipelines. Early work showed that language models can act as reward generators in purely textual settings (Kwon et al., 2023), but extending this idea to visuomotor control is nontrivial because reward specification is often ambiguous

or brittle. A natural remedy is to leverage visual reasoning to infer progress toward a goal directly from observations (Adeniji et al., 2023). One approach (Wang et al., 2024) queries a VLM to compare state images and judge improvement along a task trajectory; another aligns trajectory frames with language descriptions or demonstration captions and uses the resulting similarities as dense rewards (Fu et al., 2024). However, empirical studies indicate that such contrastive alignment introduces noise, and its reliability depends strongly on how the task is specified in language (Nam et al., 2023).

**Natural Language in Embodied AI.** With VLM architectures pushing this multimodal interface forward (Liu et al., 2023; Karamcheti et al., 2024), a growing body of work integrates visual and linguistic inputs directly into large language models to drive embodied behavior, spanning navigation (Majumdar et al., 2020), manipulation (Lynch & Sermanet, 2020), and mixed settings (Suglia et al., 2021). Beyond end-to-end conditioning, many systems focus on interpreting natural-language goals (Nair et al., 2022; Lynch et al., 2023) or on prompting strategies that extract executable guidance from an LLM—by matching generated text to admissible skills (Huang et al., 2022b), closing the loop with visual feedback (Huang et al., 2022c), incorporating affordance priors (Ahn et al., 2022), explaining observations (Wang et al., 2023b), or learning world models for prospective reasoning (Nottingham et al., 2023). Socratic Models (Zeng et al., 2022) exemplify this trend by coordinating multiple foundation models under a language interface to manipulate objects in simulation. Conversely, our framework uses natural language not as a direct policy or planner, but as structured, episodic feedback that supports causal reasoning in robotic manipulation.

**Failure Reasoning in Embodied AI.** Diagnosing and responding to failure has a long history in robotics (Khanna et al., 2023), yet many contemporary systems reduce the problem to success classification using off-the-shelf VLMs or LLMs (Ma et al., 2022; Dai et al., 2025), with some works instruction-tuning the VLM backbone to better flag errors (Du et al., 2023). Because VLMs can hallucinate or over-generalize, several studies probe or exploit model uncertainty to temper false positives (Zheng et al., 2024); nevertheless, the resulting detectors typically produce binary outcomes and provide little insight into *why* an execution failed. Iterative self-improvement pipelines offer textual critiques or intermediate feedback—via self-refinement (Madaan et al., 2023), learned critics that comment within a trajectory (Paul et al., 2023), or reflection over prior rollouts (Shinn et al., 2023)-but these methods are largely evaluated in text-world settings that mirror embodied environments, where perception and low-level control are abstracted away. In contrast, our approach targets visual robotic manipulation and treats language as structured, episodic *explanations*

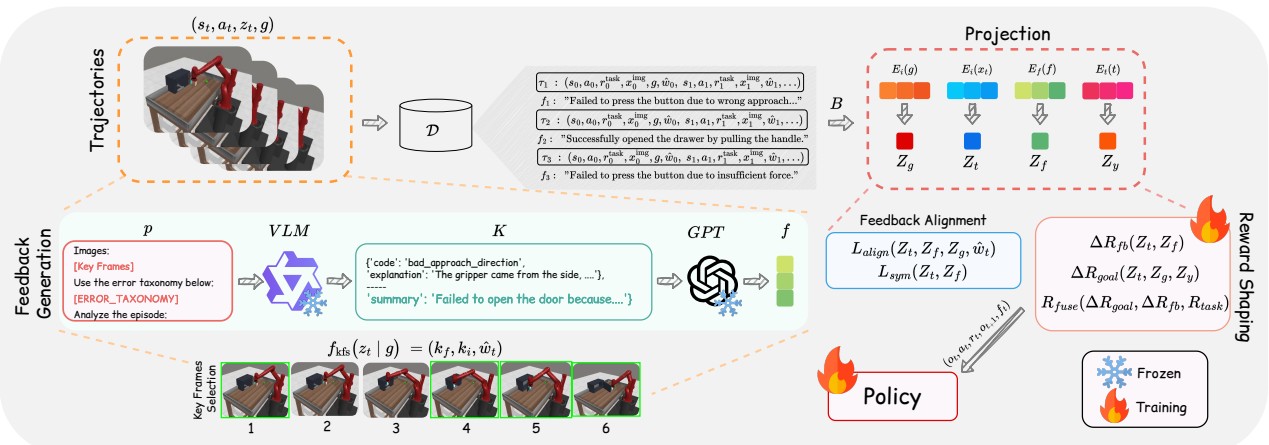

*Figure 1.* Overview of L$_A$GEA framework. **(a)** After each rollout, key–frame selection identifies causal moments and computes per-step weights $\hat{w}_t$; a VLM queried on those frames returns a schema-constrained self-reflection that is encoded as a feedback embedding $f$. Trajectories, $f$, and $\hat{w}_t$ are stored in buffer $\mathcal{D}$. **(b)** Trainable projectors ($E_i, E_t, E_f$) map state images $x_t$, goal $g$, instruction $y$, and $f$ into a shared space; a hybrid calibration+contrastive objective ($\mathcal{L}_{\text{align}}, \mathcal{L}_{\text{sym}}$) enforces control relevance. **(c)** Computes goal-delta $\Delta R_{goal}$ and feedback-delta $\Delta R_{fb}$, fuses them with sparse task reward $R_{task}$, and produces the final dense reward for policy updates.

of failure that can be aligned with image embeddings and converted into temporally grounded reward shaping signals.

## 3. Methodology

We extend on prior work (Fu et al., 2024) by incorporating a feedback-driven VLM-RL framework for embodied manipulation. Each episode, Qwen-2.5-VL-3B emits a compact, structured self-reflection, which we encode with a lightweight GPT-2 (Radford et al., 2019) model and pair it with keyframe-based saliency over the trajectory. Our framework overview is given in Figure 1.

### 3.1. Feedback Generation

To convert error-laden exploration into guidance and steer the exploration through mistakes, we employ a VLM, i.e. Qwen 2.5VL 3B (Bai et al., 2025) model for a compact, task-aware natural language reflection of what went wrong and how to proceed, which shapes subsequent learning. Appendix H, Figure 8 compactly illustrates our feedback generation pipeline.

#### 3.1.1. STRUCTURED FEEDBACK

Small VLMs can drift: the same episode rendered with minor visual differences often yields divergent, sometimes hallucinatory explanations. To make feedback reliable and comparable across training, we impose a structured protocol at the end of each episode. We uniformly sample $\mathcal{N}$ frames and prompt the VLM with the task instruction, a compact error taxonomy, two few-shot exemplars (success/failure), and a short history from the last $\mathcal{K}$ attempts. The model is required to return only a schema-constrained JSON. We

then embed the natural language episodic reflection by GPT-2, yielding a $768\text{-}D$ feedback vector that is stable across near-duplicate episodes and auditable for downstream use.

#### 3.1.2. KEY FRAME GENERATION

Uniformly broadcasting a single episodic feedback vector across all steps of the episode yields noisy credit assignment because it ignores when the outcome was actually decided. We therefore identify a small set of *key frames* and diffuse their influence locally in time, so learning focuses on causal moments (approach, contact, reversal). To keep the gate deterministic and model-agnostic, we compute key frames from the *goal-similarity trajectory* using image embeddings.

Let $x_t \in \mathbb{R}^d$ be the image embedding at time $t$ and $g \in \mathbb{R}^d$ the goal embedding. We compute a proximity signal $s_t$ and its temporal derivatives and convert them into a per-step saliency $p_t$, which favours frames that are near the goal, rapidly changing, or at sharp turns.

$$s_t = \cos(x_t, g) \in [-1, 1], \qquad v_t = s_t - s_{t-1}$$

$$a_t = v_t - v_{t-1}, \qquad v_0 = a_0 = 0$$

$$p_t = \omega_s [z(s_t)]_+ + \omega_v z(|v_t|) + \omega_a z(|a_t|); \quad \omega_s + \omega_v + \omega_a = 1$$

Here $z(\cdot)$ is a per-episode z-normalization score and $[\cdot]_+$ is ReLU. We then form $\mathcal{K}$ keyframes by selecting up to $M$ high-saliency indices with a minimum temporal spacing (endpoints always kept), yielding a compact, causally focused set of frames. We convert $\mathcal{K}$ into per-step weights with a triangular kernel (half-window $h$) and a small floor $\beta$, followed by mean normalization:

$$\tilde{w}_t = \max_{k \in \mathcal{K}} \left( 1 - \frac{|t-k|}{h+1} \right)_+, \qquad w_t = \beta + (1-\beta)\tilde{w}_t$$

These weights $\hat{w}_t$ (normalized to unit mean) concentrate mass near key frames; elsewhere, the weighting is near-uniform. They are later used in *feedback alignment*, where each timestep's contribution is scaled by $\hat{w}_t$ so image-feedback geometry is learned primarily from causal moments, and *reward shaping*, where $\hat{w}_t$ gates the per-step feedback-delta signal.

### 3.1.3. FEEDBACK ALIGNMENT

Key-frame weights $\hat{w}_t$ identify when gradients should matter; the remaining step is to make the episodic feedback $f$ actionable by aligning it with visual states in a shared space. We project images and feedback with small MLP projectors $E_i, E_f$, and use unit-norm embeddings for the image state, $z_t = \frac{E_i(x_t)}{\|E_i(x_t)\|}$, the episodic feedback $z_f = \frac{E_f(f)}{\|E_f(f)\|}$, and the goal image $z_g = \frac{E_i(g)}{\|E_i(g)\|}$. Each step is weighted by $u_t$ (key-frame saliency $\times$ goal proximity, renormalized to mean one) to concentrate updates on causal, near-goal moments.

$$\mathcal{L}_{\mathrm{bce}} = \frac{1}{\sum_t u_t} \sum_t u_t \, \mathrm{BCE}(\sigma(\psi_t/\tau_{\mathrm{bce}}), y_t)$$

$$\mathcal{L}_{\mathrm{nce}} = \frac{1}{\sum_{i:\, y_i=1} u_i} \sum_{i:\, y_i=1} u_i \, \mathrm{CE}(\mathrm{softmax}(S_{i:}), i)$$

$$\mathcal{L}_{\mathrm{align}} = \lambda_{\mathrm{bce}} \mathcal{L}_{\mathrm{bce}} + \lambda_{\mathrm{nce}} \mathcal{L}_{\mathrm{nce}}$$

Here $\psi_t = \langle z_t, z_f \rangle$, $y_t \in \{0,1\}$ and $S_{ij} = \frac{\langle z_f^{(i)}, z^{(j)} \rangle}{\tau_{\mathrm{nce}}}$.

We align feedback to vision with two complementary losses. The first enforces absolute calibration: the diagonal cosine $\psi_t = \langle z_t, z_f \rangle$ is treated as a logit (scaled by temperature $\tau_{\mathrm{bce}}$) and supervised with the per-step success label $y_t \in \{0,1\}$, so successful steps pull image and feedback together while failures push them apart. The second loss shapes the relative geometry across the batch. For each success row $i$, we form $S_{ij} = \langle z_f^{(i)}, z^{(j)} \rangle / \tau_{\mathrm{nce}}$ and apply cross-entropy over columns so feedback $i$ prefers its own image over batch negatives. The hybrid objective balances these terms via hyperparameters $\lambda_{\mathrm{bce}}, \lambda_{\mathrm{nce}}$.

To further polish the geometry, we refine the shared space with a symmetric, weighted contrastive step that uses the same weights but averages the cross-entropy in both directions (feedback-to-image and image-to-feedback). With per-row weights renormalized, label smoothing, and small regularizers ($\lambda_{align}, \lambda_{uni}$) for pairwise alignment and uniformity on the unit sphere, the update becomes,

$$\mathcal{L}_{\mathrm{sym}} = \frac{1}{2}\big[\mathrm{CE}_{fi} + \mathrm{CE}_{if}\big] + \lambda_{\mathrm{align}} \, \mathbb{E}\|z_t^{(i)} - z_f^{(i)}\|^2$$
$$+ \lambda_{\mathrm{uni}} \, \log \mathbb{E}_{\substack{a \neq b \\ z_a, z_b \in \mathcal{Z}}} \exp\big(-2\|z_a - z_b\|^2\big)$$

Here, $\mathrm{CE}_{fi}$ and $\mathrm{CE}_{if}$ are cross-entropies over cosine-similarity softmaxes from feedback to image and image to

to feedback, and $a, b$ index distinct unit–norm embeddings $z_a, z_b \in \mathcal{Z}$ from the current minibatch (images and feedback).

Together, the calibration (BCE), discrimination (InfoNCE) (Oord et al., 2018), and symmetric refinement yield a stable, control-relevant geometry driven by key frames near the goal. Key-frame and goal-proximity weights ensure these gradients come from moments that matter. The learned projector is used downstream to compute goal and feedback-delta potentials for reward shaping, and to estimate instruction text–feedback agreement for reward fusion.

## 3.2. Reward Generation

With the shared space in place, we convert progress toward the task and movement toward the feedback into dense, directional rewards. We project images, instruction text, and feedback with $E_i, E_t, E_f$ and use unit–norm embeddings for the current state $z_t$, the goal image $z_g$, the episodic feedback $z_f$, and the instruction text $z_y = \frac{E_t(\mathrm{instruction})}{\|E_t(\mathrm{instruction})\|}$. Potentials are squashed with $\tanh$ to keep scale bounded and numerically stable. We define a goal potential $\phi_t$ by averaging instruction text- and image-goal affinities, then shape its temporal difference and get the goal-delta reward, $r_t^{goal}$:

$$\phi_t = \frac{1}{2}\Big[\tanh\Big(\frac{0.5(\langle z_t, z_y \rangle + 1) - 0.5}{\tau_{\mathrm{goal}}}\Big)$$
$$+ \tanh\Big(\frac{0.5(\langle z_t, z_g \rangle + 1) - 0.5}{\tau_{\mathrm{goal}}}\Big)\Big]$$
$$r_t^{\mathrm{goal}} = \tanh\Big(\frac{\gamma\,\phi_{t+1} - \phi_t}{\tau_{\mathrm{goal}}}\Big)$$

where $\gamma \in (0,1)$ is the shaping discount and $\tau_{\mathrm{goal}} > 0$ controls slope. $r_t^{goal}$ supplies shaped progress signals while preserving scale, and is positive when the state moves closer to the goal and negative otherwise.

In parallel, we reward movement toward the feedback direction and concentrate credit to causal moments via the key–frame weights $\hat{w}_t$. Let $\psi_t = \langle z_t, z_f \rangle$ be feedback embeddings cosine with the state and feedback temperature $\tau_f > 0$ shaping the slope, we form a feedback-delta reward, $r_t^{fb}$. We then combine goal and feedback delta reward and get the fused reward $\tilde{r}_t$ using a confidence–aware mixture that increases with instruction–feedback agreement, $a = \frac{1}{2}(1 + \langle z_y, z_f \rangle) \in [0,1]$

$$r_t^{\mathrm{fb}} = \hat{w}_t \, \tanh\Big(\frac{\gamma\,\psi_{t+1} - \psi_t}{\tau_{\mathrm{f}}}\Big), \quad \psi_t = \langle z_t, z_f \rangle$$

$$\tilde{r}_t = (1-\alpha)\, r_t^{\mathrm{goal}} + \alpha\, r_t^{\mathrm{fb}}; \quad \alpha = \mathrm{clip}\big(\alpha_{\mathrm{base}} \cdot a, \, [\alpha_{\mathrm{min}}, \alpha_{\mathrm{max}}]\big)$$

Here, $\alpha_{\mathrm{base}}, \alpha_{\mathrm{min}}, \alpha_{\mathrm{max}}$ are hyperparameters. All terms are $\tanh$-bounded, so $\tilde{r}_t \in [-1,1]$, providing informative reward signals without destabilizing the critic. In the next subsection we describe how $\tilde{r}_t$ is added to the environment task reward $r_t^{\mathrm{task}} \in \{-1,1\}$ under an adaptive $\rho$-schedule.

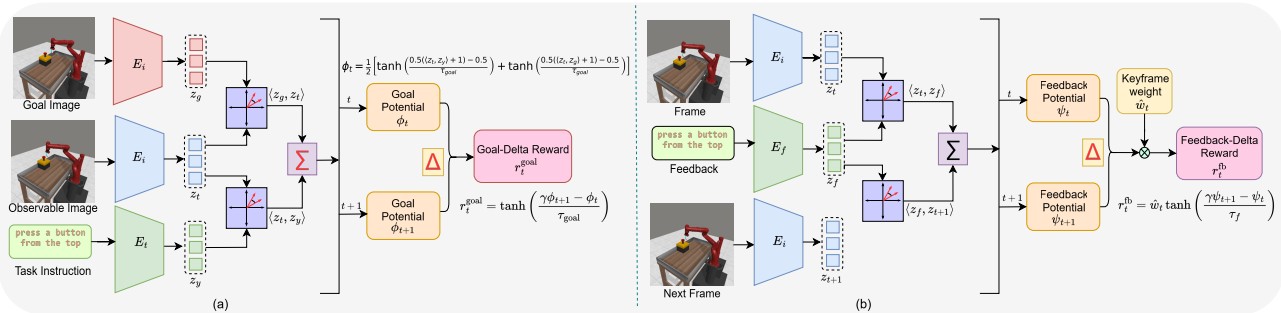

*Figure 2.* The computation of our delta-based rewards. (a) A Goal Potential $\phi_t$ is formed by aligning the current state $z_t$ with the goal image $z_g$ and instruction $z_y$. (b) A Feedback Potential $\psi_t$ is formed by aligning $z_t$ with the VLM feedback $z_f$. The temporal difference of these potentials creates the fused feedback-VLM rewards.

## 3.3. Dynamic Reward Shaping

Critic receives, reward signal $r = r_t^{\text{task}} + \rho \, \tilde{r}_t$, where $r_t^{\text{task}}$ is the environment task reward. Environment task reward $r_t^{\text{task}}$ is episodic and sparse, whereas the fused VLM signal $\tilde{r}_t$ is dense but can overpower the task reward if used naively. We therefore gate shaping with a coefficient $\rho$, that is failure-focused, progress-aware, and smooth, so language guidance is strong when exploration needs direction and recedes as competence emerges.

We apply shaping only on failures using the mask $m_t = \mathbf{1}[\, r_t^{\text{task}} < 0\,]$, and we down-weight shaping as the policy improves. Progress is estimated in $\bar{s} \in [0, 1]$ by combining an episodic success exponential moving average (EMA) with a batch-level improvement signal from the goal delta.

$$P = \max\Big(\bar{s}, \, \big(\tfrac{1}{B}\sum_t \mathbf{1}[\, r_t^{\text{goal}} > 0\,]\big)^2\Big).$$

$$\rho_t = \rho_{\min} + (\rho_{\max} - \rho_{\min})\,(1 - P);\, 0 < \rho_{\min} < \rho_{\max} < 1$$

We map $P$ to an effective shaping weight $\rho_t$, so that shaping is large early and fades as competence grows. As the shaping is only applied to failures $m_t$, per-step shaped coefficient becomes $\hat{\rho}_t = m_t \rho_t$. The SAC algorithm is finally trained on, reward $r_t = r_t^{\text{task}} + \hat{\rho}_t \tilde{r}_t$, which preserves the task reward while letting VLM shaping accelerate exploration and early credit assignment, then gradually relinquish control as the policy becomes competent. The pseudo-code algorithm of LᴀGEA is illustrated in the Appendix G.

## 4. Experiments

We evaluate LᴀGEA on a suite of simulated embodied manipulation tasks, comparing against baseline RL agents and ablated LᴀGEA variants to measure the contributions of VLM-driven self-reflection, keyframes selection, and feedback alignment. Our experiments demonstrate that incorporating compact, structured feedback from VLM's leads to faster learning, more robust policies, and improved general-

ization to goal configurations. We investigate the following research questions:

**RQ1:** How much does VLM-guided feedback improve policy learning and task success?

**RQ2:** Does natural language feedback guide embodied agents to achieve policy convergence faster?

**RQ3:** How robust is the design of LᴀGEA?

**Setup:** We evaluate LᴀGEA framework on ten robotics tasks from the Meta-world MT10 benchmark (Yu et al., 2020) and Robotic Fetch (Plappert et al., 2018), utilizing sparse rewards. LᴀGEA leverages Qwen-2.5-VL-3B for generating structured feedback, encoded with GPT-2. Visual observations are embedded using the LIV model (Ma et al., 2023). Implementation details are available in Appendix F.

## 4.1. RQ1: How much does VLM-guided feedback improve policy learning and task success?

**Baseline:** To thoroughly evaluate LᴀGEA, we compare its performance against a suite of relevant reward learning baselines. We begin with a standard Soft Actor-Critic (SAC) agent (Haarnoja et al., 2018) trained solely on the sparse binary task reward. We also include LIV (Ma et al., 2023), a robotics reward model pre-trained on large-scale datasets, and a variant, LIV-Proj, which utilizes randomly initialized and fixed projection heads for image and language embeddings. To further assess the benefits of exploration strategies, we incorporate Relay (Lan et al., 2023), a simplified approach that integrates relay RL into the LIV baseline. Finally, we compare against FuRL (Fu et al., 2024), a method employing reward alignment and relay RL to address fuzzy VLM rewards.

### 4.1.1. Rᴇsᴜʟᴛs ᴏɴ Mᴇᴛᴀᴡᴏʀʟᴅ MT10

Our experiments on the Meta-World MT10 benchmark demonstrate the effectiveness of LᴀGEA in leveraging

*Table 1.* Experiment results on MT10 benchmarks with fixed goal. Average success rate across five random seeds.

| Environment | SAC | LIV | LIV-Proj | Relay | FuRL w/o goal-image | FuRL | LᴀGEA |
|---|---|---|---|---|---|---|---|
| $r^{\mathrm{VLM}}feed$ | ✗ | ✗ | ✗ | ✗ | ✗ | ✗ | ✓ |
| $r^{\mathrm{VLM}}$ | ✗ | ✓ | ✓ | ✓ | ✓ | ✓ | ✓ |
| $r^{\mathrm{task}}$ | ✓ | ✓ | ✓ | ✓ | ✓ | ✓ | ✓ |
| button-press-topdown-v2 | 0 | 0 | 0 | 60 | 80 | 100 | 100 |
| door-open-v2 | 50 | 0 | 0 | 80 | 100 | 100 | 100 |
| drawer-close-v2 | 100 | 100 | 100 | 100 | 100 | 100 | 100 |
| drawer-open-v2 | 20 | 0 | 0 | 40 | 80 | 80 | 100 |
| peg-insert-side-v2 | 0 | 0 | 0 | 0 | 0 | 0 | 0 |
| pick-place-v2 | 0 | 0 | 0 | 0 | 0 | 0 | 0 |
| push-v2 | 0 | 0 | 0 | 0 | 40 | 80 | 100 |
| reach-v2 | 60 | 80 | 80 | 100 | 100 | 100 | 100 |
| window-close-v2 | 60 | 60 | 40 | 80 | 100 | 100 | 100 |
| window-open-v2 | 80 | 40 | 20 | 80 | 100 | 100 | 100 |
| Average | 37.0 | 28.0 | 24.0 | 54.0 | 70.0 | 76.0 | **80.0** |

| Task | SAC | Relay | FuRL | LᴀGEA |
|---|---|---|---|---|
| button-press-topdown-v2 | 16.0 (32.0) | 56.0 (38.3) | 64.0 (32.6) | 96 (8) |
| door-open-v2 | 78.0 (39.2) | 80.0 (30.3) | 96.0 (8.0) | 100 (0) |
| drawer-close-v2 | 100.0 (0.0) | 100.0 (0.0) | 100.0 (0.0) | 100 (0) |
| drawer-open-v2 | 40.0 (49.0) | 50.0 (42.0) | 84.0 (27.3) | 92 (9.8) |
| pick-place-v2 | 0.0 (0.0) | 0.0 (0.0) | 0.0 (0.0) | 4 (4.9) |
| peg-insert-side-v2 | 0.0 (0.0) | 0.0 (0.0) | 0.0 (0.0) | 0 |
| push-v2 | 0.0 (0.0) | 0.0 (0.0) | 6.0 (8.0) | 12 (4) |
| reach-v2 | 100.0 (0.0) | 100.0 (0.0) | 100.0 (0.0) | 100 (0) |
| window-close-v2 | 86.0 (28.0) | 96.0 (4.9) | 100.0 (0.0) | 100 (0) |
| window-open-v2 | 78.0 (39.2) | 92.0 (7.5) | 96.0 (4.9) | 100 (0) |
| Average | 49.8 (7.9) | 57.4 (7.0) | 64.6 (5.0) | **70.4 (1.85)** |

*Table 2.* Experiment results on MT10 benchmarks with random goal. We present the average success rate across five random seeds.

| Task | SAC | Relay | FuRL | LᴀGEA |
|---|---|---|---|---|
| Reach-v2 | 100 (0) | 100 (0) | 100 (0) | 100 (0) |
| Push-v2 | 26.67 (4.71) | 30 (8.16) | 40 (8.16) | 53.33 (4.71) |
| PickAndPlace-v2 | 10 (8.16) | 20 (0) | 33.33 (9.43) | 43.33 (4.71) |
| Slide-v2 | 0 (0) | 0 (0) | 3.33 (4.71) | 10 (8.16) |
| Average | 34.17 | 37.5 | 44.17 | **51.67** |

*Table 3.* Experiment results on Fetch manipulation suite. Average success rate (STD) across three different seeds; higher is better.

**VLM feedback for reinforcement learning.** As shown in Table 1, LᴀGEA achieves a strong performance improvement of 5.3% over baselines, with an average success rate of 80% on hidden-fixed goal tasks. More importantly, its true strength lies in its ability to generalize to varied goal positions. In the observable-random goal setting (Table 2), LᴀGEA achieves a 70.4% average success rate, representing a 9% improvement over all baselines. While FuRL achieves respectable performance, LᴀGEA consistently surpasses it in the hidden-fixed goal setting as well as tasks in the more challenging observable-random goal setting.

#### 4.1.2. RESULTS ON FETCH TASKS

We further evaluate LᴀGEA on the Robotic Fetch (Plappert et al., 2018) manipulation suite to assess its effectiveness in sparse-reward, goal-conditioned control. As summarized in Table 3, we report the average success rate where LᴀGEA consistently outperforms all baselines across the four Fetch tasks. While SAC struggles with sparse supervision (34.17%), and Relay and FuRL provide moderate improvements (37.5% and 44.17%), LᴀGEA achieves the highest average success rate of 51.67%, representing a 17% absolute improvement over the strongest baseline.

### 4.2. RQ2: Does natural language feedback guide embodied agents to achieve policy convergence faster?

Figure 3 provides a comprehensive comparison of convergence dynamics across eight Meta-World tasks, offering a definitive answer to our research question (RQ2). The results demonstrate that LᴀGEA achieves significantly faster policy convergence than both the FuRL and SAC baselines in almost all of the tasks. The efficiency of LᴀGEA is evident, as it consistently reaches task completion substantially sooner than its counterparts. This accelerated learning is driven by the dense, corrective signals from our feedback mechanism, which fosters a more effective exploration process compared to the slower, incremental learning of FuRL or the near-complete failure of sparse-reward SAC. Even on the most challenging tasks (button-press-topdown-v2 and drawer-open-v2), LᴀGEA is the only method to show meaningful, non-zero success, demonstrating its ability to provide actionable guidance where other methods fail.

### 4.3. RQ3: How robust is the design of LᴀGEA?

To validate our design choices and disentangle the individual contributions of our core components, we conduct a series of comprehensive ablation studies. Our analysis focuses on the primary modules of the LᴀGEA framework: (1) Reward Engineering ( 4.3.1), which includes the delta reward formulation and the dynamic reward shaping schedule; (2) Keyframe Selection mechanism ( 4.3.2), designed to solve the feedback credit assignment problem; (3) Feedback Quality ( 4.3.3), to determine the usefulness of structured vs free-form feedback, (4) Feedback Alignment module ( 4.3.4) responsible for creating a control-relevant embedding space,

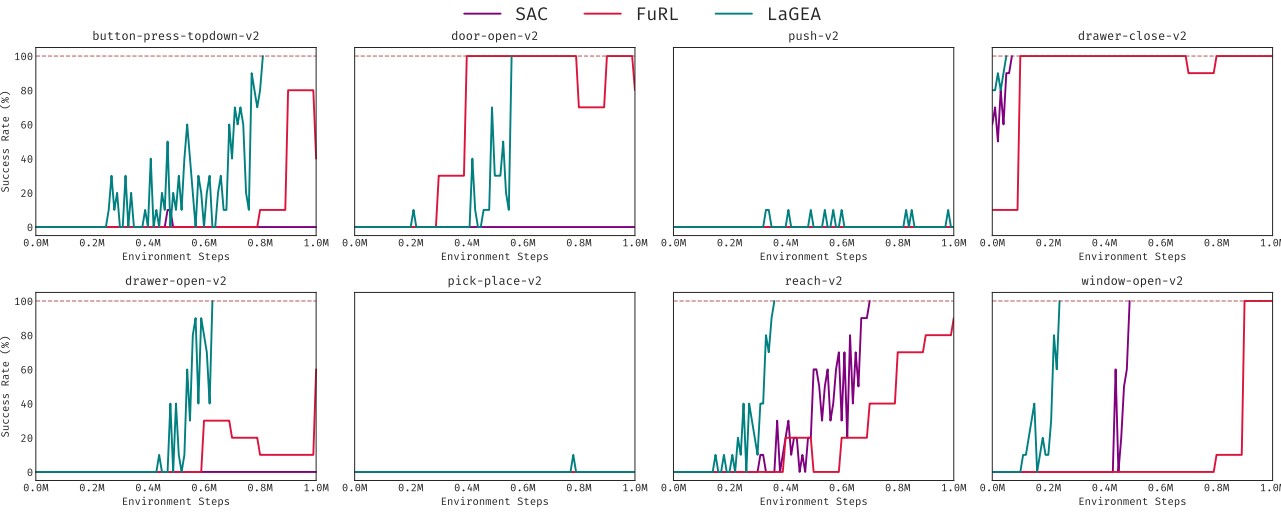

*Figure 3.* Natural-language feedback accelerates convergence: across eight Meta-World tasks, LᴀGEA reaches high success in far fewer steps than FuRL and SAC, which plateau late or stall.

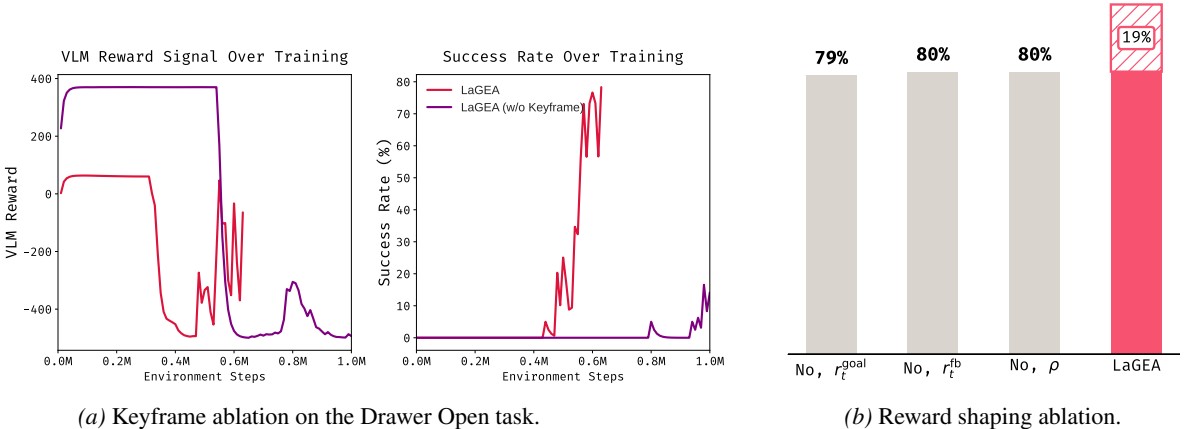

*(a)* Keyframe ablation on the Drawer Open task.      *(b)* Reward shaping ablation.

*Figure 4.* Ablation studies on keyframe selection and reward shaping.

and finally (5) viewpoint-shift experiment ( 4.3.5) to test robustness to camera changes. Our central finding is that these components are highly synergistic; while each provides a significant contribution, the full performance of LᴀGEA is only realized through their combined effort.

### 4.3.1. Sʏɴᴇʀɢʏ ᴏꜰ Dᴇʟᴛᴀ Rᴇᴡᴀʀᴅꜱ ᴀɴᴅ Aᴅᴀᴘᴛɪᴠᴇ Sʜᴀᴘɪɴɢ

To isolate the contributions of our key reward components, we performed a targeted ablation study on both observable random goal and hidden fixed goal tasks (e.g., button press topdown, drawer open, door open), with results visualized in Figure 4b. This analysis demonstrates the roles of goal delta reward, $r_t^{goal}$, feedback delta reward, $r_t^{fb}$ and our proposed dynamic reward shaping, $\rho$. Figure 4b un-

equivocally demonstrates that all components are critical and contribute synergistically to the high performance of the full LᴀGEA system. The complete LᴀGEA framework achieves a near-perfect average success score outperforming other baselines in these experiments. In contrast, removing any single component leads to a substantial performance degradation. This assesment suggest that the components of our reward generation are not merely additive but deeply complementary. As visualized in the Figure 4b, the final 19% performance gain achieved by the full LᴀGEA model over the best-performing ablation is a direct result of the synergy between measuring long-term progress, incorporating short-term corrective feedback, and dynamically balancing this guidance as the agent's competence grows.

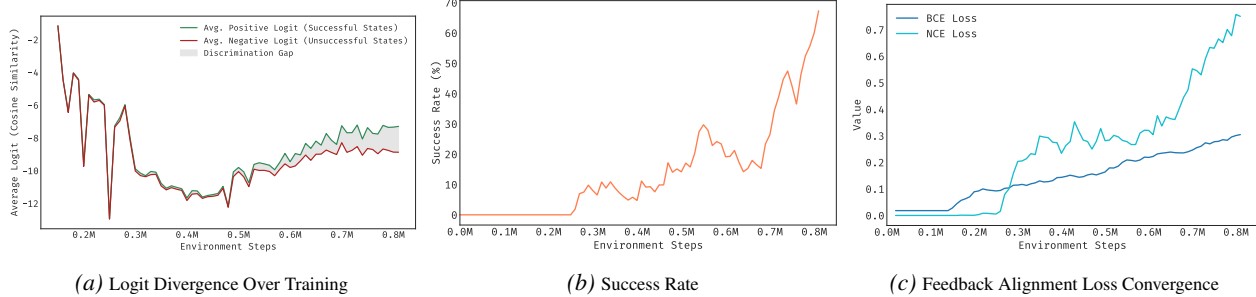

*(a)* Logit Divergence Over Training  *(b)* Success Rate  *(c)* Feedback Alignment Loss Convergence

*Figure 5.* Alignment enables control-relevant geometry: (a) success/failure logit margin increases over training, (b) policy success accelerates, and (c) BCE/InfoNCE objectives co-train the shared space for LАGEA.

| Task | Random KF (STD) | Uniform KF (STD) | LaGEA (STD) |
|---|---|---|---|
| Button press | 36.67 (17) | 50 (24.49) | 96 (8) |
| Door open | 93.33 (9.43) | 100 (0) | 100 (0) |
| Drawer open | 100 (0) | 80 (16.33) | 92 (9.8) |
| Push | 10 (0) | 6.67 (9.43) | 12 (4) |
| Window open | 100 (0) | 100 (0) | 100 (0) |
| Average | 68 | 67.33 | **80** |

*Table 4.* Comparison among randomly sampled, uniformly sampled, and LaGEA sampled keyframes for five Meta-World observable tasks on three random seeds.

| Task | Freeform Feedback | Structured Feedback |
|---|---|---|
| Button press topdown v2 obs. | 10 | 93.33 |
| Drawer open v2 obs. | 96.67 | 100 |
| Door open v2 obs. | 100 | 100 |
| Push v2 hidden | 66.67 | 100 |
| Drawer open v2 hidden | 100 | 100 |
| Door open v2 hidden | 100 | 100 |
| Average | 78.89 | **98.89** |

*Table 5.* Average performance of Freeform vs Structured Feedback across three different seeds.

### 4.3.2. KEYFRAME EXTRACTION & CREDIT ASSIGNMENT

Figure 4a visualizes the ablation on the *Drawer Open* task, showing the impact of our keyframe generation mechanism. LАGEA with keyframing learns the task efficiently, while the variant without keyframing catastrophically fails. As the agent learns to approach the goal correctly, the VLM reward signal appropriately increases, reflecting true progress just before the agent achieves success. We also conducted a direct keyframe comparison on Table 4, across randomly sampled frames, uniformly sampled frames, and LaGEA keyframes on the same complicated MetaWorld five observable tasks as Appendix C. LАGEA outperforms both random and uniformly-spaced keyframes on average. Our selector is stronger because it concentrates credit around causal moments derived from the goal-similarity trajectory, rather than spreading the signal uniformly or stochastically. The agent, without keyframing, lacks focused guidance and fails to make crucial connections and thus remains trapped in its suboptimal policy.

### 4.3.3. IMPACT OF STRUCTURED FEEDBACK

We conducted a crucial study comparing our structured feedback approach against a baseline using free-form textual feedback from the VLM to validate our hypothesis regarding the benefits of structured VLM feedback. The results, presented in Table 5, show a clear and significant advantage for using structured feedback. On average, our structured feedback approach outperforms the freeform feedback baseline.

We attribute this performance disparity to feedback consistency. Freeform feedback, while expressive, introduces significant challenges by generating verbose, ambiguous, or irrelevant text, leading to noisy and often misleading guidance. In contrast, our structured taxonomy compels the VLM to provide a compact, unambiguous, and consistently formatted signal, which enables reliable guidance.

### 4.3.4. FEEDBACK-REWARD ALIGNMENT

To provide a deeper insight into our framework, we visualize the interplay between agent performance and the internal metrics of our feedback alignment module in Figure 5. The plots illustrate a clear, causal relationship: successful policy learning is contingent upon the convergence of a meaningful, control-relevant embedding space as engineered by our methodology. Initially, as shown in Figure 5a, the average logits for successful and unsuccessful states $\psi_t = \langle z_t, z_f \rangle$ are alike. This indicates that our hybrid alignment objective, $\mathcal{L}_{\text{align}}$, has not yet converged, and the feedback is not yet meaningfully aligned with the visual states. Consequently, the agent's success rate remains at zero (Figure 5b). The turning point occurs around the 0.5M step mark, where a stable and growing *Discrimination Gap* emerges. This is direct evidence of our methodology at work: the $\mathcal{L}_{\text{bce}}$ component is successfully calibrating the logits based on the success label $y_t$, while the contrastive $\mathcal{L}_{\text{nce}}$ term is simultaneously shaping the relative geometry to distinguish correct pairs from negatives within the batch. Figure 5c reveals the cause of this emergent structure: as the agent's policy improves, it

*Table 6.* LᴀGEA is trained from the top-left diagonal front camera viewpoint and evaluated under three unseen camera viewpoints. Values in parentheses denote standard deviations over three random seeds.

| Task | Directly overhead | Front-left diagonal | Behind-left diagonal | LaGEA |
|------|-------------------|---------------------|----------------------|-------|
| Button press topdown v2 obs. | 83.33 (12.47) | 80 (16.33) | 83.33 (17) | 96 (8) |
| Door open v2 obs. | 100 (0) | 100 (0) | 100 (0) | 100 (0) |
| Drawer open v2 obs. | 100 (0) | 100 (0) | 100 (0) | 92 (9.8) |
| Push v2 obs. | 16.67 (4.71) | 10 (0) | 13.33 (4.71) | 12 (4) |
| Window open v2 obs. | 96.67 (4.71) | 96.67 (4.71) | 100 (0) | 100 (0) |
| Average | 79.33 | 77.33 | 79.33 | **80** |

presents the alignment module with more challenging hard negative trajectories, causing the BCE and NCE losses to rise. This rising loss is not a sign of failure but a reflection of a co-adaptive learning process where the alignment module is forced to learn the fine-grained distinctions.

### 4.3.5. Vɪᴇᴡᴘᴏɪɴᴛ Rᴏʙᴜsᴛɴᴇss

Although LᴀGEA doesn't claim to solve severe self-occlusion or cluttered-scene occlusion, it is pretty robust to viewpoint shifts and camera changes. LᴀGEA is trained entirely at viewpoint 2 (Top-Left diagonal-front camera) and evaluated zero-shot at three held-out viewpoints: directly overhead, looking straight down; front-left diagonal, slightly elevated; and behind-left diagonal, elevated top-view. These four viewpoints represent meaningfully different spatial perspectives. We utilized the five similar observation-based MT10 tasks already used in the Appendix C. We chose this subset because it is already the paper's observation-based robustness subset, and because it covers a compact but varied range of manipulation types. As illustrated in Table 6 LᴀGEA sustains strong performance across all three unseen viewpoints (77%–79% average), with negligible variance between them, although severe occlusion remains an open limitation.

We further conduct additional experiments, such as the impact of different VLMs/Text Encoders on observation-based manipulation tasks. More in Appendix C

## 5. Conclusion

Natural-language can be a training signal as error feedback for embodied manipulation rather than mere goal description. We present LᴀGEA, which operationalizes this idea by turning schema-constrained episodic reflections into temporally grounded reward shaping through keyframe-centric gating, feedback-vision alignment, and an adaptive, failure-aware representation. On the Meta-World MT10 and Robotic Fetch benchmark, LᴀGEA improves average success over SOTA by a large margin with faster convergence, substantiating our claim that time-grounded language feedback sharpens credit assignment and exploration, enabling

agents to learn from mistakes more effectively.

LᴀGEA still inherits occasional hallucinations from the underlying VLM, which our structure and alignment mitigate but cannot eliminate. While the study spans diverse simulated tasks, real-robot generalization and long-horizon observability remain open challenges. A natural next step is to translate from simulation to real-robot deployment, closing the sim-to-real gap.

## Impact Statement

This paper introduces LᴀGEA, a framework that converts structured reflections from a vision language model into temporally grounded reward shaping for reinforcement learning in robotic manipulation. By reducing reliance on hand-crafted rewards and improving learning efficiency in sparse feedback settings, this approach could lower the cost of developing manipulation skills for assistive robotics, flexible automation, and faster scientific iteration. LᴀGEA takes one of the first steps towards utilizing natural language to learn from mistakes. Over time, in the broader field of Embodied AI, we think this work will contribute significantly to developing generalist robots that can perceive and learn from the environment.

Regardless, as this framework was experimented on in simulation, and no real-world subjects were impacted while developing this, we believe that the overall project did not violate any ethical concerns. However, over the long horizon, when expanding this work from sim-to-real, there could be some potential societal consequences of our work. The size of these effects is uncertain, especially when moving from simulation to real-world deployment; nevertheless, we perceive them to be beyond the scope of our work to be considered here.

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

## A. LLM Usage

We used ChatGPT (GPT-5 Thinking) solely as a general-purpose writing assistant to refine prose after complete, author-written drafts were produced. Its role was limited to language editing, i.e. suggesting alternative phrasings, improving clarity and flow, and reducing redundancy without introducing new citations or technical claims. The research idea, methodology, experiments, analyses, figures, and all substantive content were conceived and executed by the authors. LLMs were not used for ideation, data analysis, or result generation. All AI-assisted text was reviewed, verified, and, when necessary, rewritten by the authors, who take full responsibility for the manuscript's accuracy and originality.

## B. Extended Related Work

**VLMs for RL.** Foundation models (Wiggins & Tejani, 2022) have proven broadly useful across downstream applications (Ramesh et al., 2022; Khandelwal et al., 2022; Chowdhury et al., 2025), motivating their incorporation into reinforcement learning pipelines. Early work showed that language models can act as reward generators in purely textual settings (Kwon et al., 2023), but extending this idea to visuomotor control is nontrivial because reward specification is often ambiguous or brittle. A natural remedy is to leverage visual reasoning to infer progress toward a goal directly from observations (Mahmoudieh et al., 2022; Rocamonde et al., 2023; Adeniji et al., 2023). One approach (Wang et al., 2024) queries a VLM to compare state images and judge improvement along a task trajectory; another aligns trajectory frames with language descriptions or demonstration captions and uses the resulting similarities as dense rewards (Fu et al., 2024; Rocamonde et al., 2023). However, empirical studies indicate that such contrastive alignment introduces noise, and its reliability depends strongly on how the task is specified in language (Sontakke et al., 2023; Nam et al., 2023).

**Natural Language in Embodied AI.** With VLM architectures pushing this multimodal interface forward (Liu et al., 2023; Karamcheti et al., 2024; Laurençon et al., 2024), a growing body of work integrates visual and linguistic inputs directly into large language models to drive embodied behavior, spanning navigation (Fried et al., 2018; Wang et al., 2019; Majumdar et al., 2020), manipulation (Lynch & Sermanet, 2020), and mixed settings (Suglia et al., 2021; Fu et al., 2019; Hill et al., 2020). Beyond end-to-end conditioning, many systems focus on interpreting natural-language goals (Lynch & Sermanet, 2020; Nair et al., 2022; Shridhar et al., 2022; Lynch et al., 2023) or on prompting strategies that extract executable guidance from an LLM—by matching generated text to admissible skills (Huang et al., 2022b), closing the loop with visual feedback (Huang et al., 2022c), planning over maps or graphs (Shah et al., 2023; Huang et al., 2022a), incorporating affordance priors (Ahn et al., 2022), explaining observations (Wang et al., 2023b), learning world models for prospective reasoning (Nottingham et al., 2023; Zellers et al., 2021), or emitting programs and structured action plans (Liang et al., 2022; Singh et al., 2022). Socratic Models (Zeng et al., 2022) exemplify this trend by coordinating multiple foundation models (e.g., GPT-3 (Brown et al., 2020) and ViLD (Gu et al., 2021)) under a language interface to manipulate objects in simulation. Conversely, our framework uses natural language not as a direct policy or planner, but as structured, episodic feedback that supports causal credit assignment in robotic manipulation.

**Failure Reasoning in Embodied AI.** Diagnosing and responding to failure has a long history in robotics (Ye et al., 2019; Khanna et al., 2023), yet many contemporary systems reduce the problem to success classification using off-the-shelf VLMs or LLMs (Ma et al., 2022; Ha et al., 2023; Wang et al., 2023a; Duan et al., 2024; Dai et al., 2025), with some works instruction-tuning the vision–language backbone to better flag errors (Du et al., 2023). Because large models can hallucinate or over-generalize, several studies probe or exploit model uncertainty to temper false positives (Zheng et al., 2024); nevertheless, the resulting detectors typically produce binary outcomes and provide little insight into *why* an execution failed. Iterative self-improvement pipelines offer textual critiques or intermediate feedback—via self-refinement (Madaan et al., 2023), learned critics that comment within a trajectory (Paul et al., 2023), or reflection over prior rollouts (Shinn et al., 2023)-but these methods are largely evaluated in text-world settings that mirror embodied environments such as ALFWorld (Shridhar et al., 2020), where perception and low-level control are abstracted away. In contrast, our approach targets visual robotic manipulation and treats language as structured, episodic *explanations* of failure that can be aligned with image embeddings and converted into temporally grounded reward shaping signals.

## C. Additional Experimental Results

### C.1. What happens with different VLMs/Encoders?

Table 7 presents results on observation-based manipulation tasks using different VLM backbones, including SmolVLM2 (Marafioti et al., 2025), InternVL2 (Chen et al., 2024c), and OpenQwen2VL (Wang et al., 2025). The

| Task | SmolVLM2 | InternVL2 | OpenQwen2VL | LᴀGEA |
|---|---|---|---|---|
| Button-press-topdown-v2-observable | 20 (28.28) | 56.67 (24.94) | 40 (37.42) | 93.33 (9.43) |
| Door-open-v2-observable | 56.67 (36.82) | 100 (0) | 100 (0) | 100 (0) |
| Drawer-open-v2-observable | 93.33 (9.43) | 90 (14.14) | 93.33 (9.43) | 93.33 (9.43) |
| Push-v2-observable | 10 (0) | 3.33 (4.71) | 0 (0) | 13.33 (4.71) |
| Window-open-v2-observable | 100 (0) | 90 (8.16) | 100 (0) | 100 (0) |
| Average | 56 | 68 | 66.67 | **80** |

*Table 7.* Effect of using different VLMs on task success. Results are averaged over three random seeds; higher is better.

| Task | LIV | BGE | MPNet | LaGEA |
|---|---|---|---|---|
| Button-press-topdown-v2-observable | 63.33 (4.71) | 56.67 (41.9) | 50 (28.28) | 93.33 (9.43) |
| Door-open-v2-observable | 100 (0) | 100 (0) | 100 (0) | 100 (0) |
| Drawer-open-v2-observable | 96.67 (4.71) | 100 (0) | 96.67 (4.71) | 93.33 (9.43) |
| Push-v2-observable | 3.33 (4.71) | 0 (0) | 6.67 (4.71) | 13.33 (4.71) |
| Window-open-v2-observable | 100 (0) | 100 (0) | 100 (0) | 100 (0) |
| Average | 72.67 | 71.33 | 70.67 | **80** |

*Table 8.* Effect of different text encoders on observation-based manipulation tasks. Results are averaged over three random seeds (Standard Deviation is in brackets); higher is better.

comparison highlights that while stronger VLM backbones improve performance overall, LᴀGEA achieves the highest average success rate, indicating that its feedback grounding and reward shaping strategy is robust across model choices.

Additionally, we study the impact of different text encoders, including LIV (Ma et al., 2023), BGE (Chen et al., 2024a), and MPNet (Song et al., 2020) as shown in Table 8. For different VLMs/LLMs, LᴀGEA almost always performs significantly better. Therefore, LᴀGEA pipeline is reasonably robust to the choice of VLM/LLM encoders. All combinations outperform vanilla SAC or FuRL, and several combinations are close to our default; Qwen2.5-VL-3B + GPT-2 simply offers the best average performance, which is why we use it in the main results.

## C.2. Wall-Clock Time to Convergence

| Task | FuRL (min, STD) | LᴀGEA(min, STD) |
|---|---|---|
| Drawer-close-v2-observable | 21.51 (3.30) | 18.37 (9.42) |
| Window-close-v2-observable | 165.84 (38.06) | 98.78 (14.51) |
| Reach-v2-observable | 183.64 (23.00) | 157.68 (37.45) |
| Button-press-topdown-v2-hidden | 99.71 (36.93) | 162.12 (17.71) |
| Door-open-v2-hidden | 88.03 (30.57) | 86.90 (23.26) |
| Drawer-close-v2-hidden | 14.89 (2.02) | 14.49 (4.71) |
| Reach-v2-hidden | 43.62 (7.26) | 60.11 (11.57) |
| Window-close-v2-hidden | 108.31 (52.13) | 123.85 (26.70) |
| Window-open-v2-hidden | 128.59 (23.61) | 108.90 (46.37) |
| **Average** | 94.90 | **92.36** |

*Table 9.* Wall-clock time to convergence (in minutes) for FuRL and LᴀGEAon Meta-World MT10 tasks. Results are averaged over three seeds with standard deviation in parentheses.

Table 9 compares the wall-clock time to convergence between FuRL and LᴀGEA across Meta-World MT10 tasks. Across these tasks, LᴀGEA reaches convergence slightly faster on average (92.36 vs. 94.90 minutes) despite the extra cost of VLM inference. This is because our framework typically requires fewer environment steps to solve a task than FuRL, as also visible in the learning curves in Figure 3, so the additional reflection cost is compensated by improved sample efficiency.

# D. Experimental Setup

All experiments (including ablations) were run on a Linux workstation running Ubuntu 24.04.2 LTS (kernel 6.14.0-29-generic). The machine is equipped with an Intel Core Ultra 9 285K CPU, 96 GB of system RAM, and an NVIDIA GeForce RTX 4090 (AD102, 24 GB VRAM) serving as the primary accelerator; an integrated Arrow Lake-U graphics adapter is present but unused for training. Storage is provided by a 2 TB NVMe SSD (MSI M570 Pro). The NVIDIA proprietary driver was used for the RTX 4090, and all training/evaluation leveraged GPU acceleration; results reported in the paper were averaged over multiple random seeds with identical software and driver configurations on this host.

# E. Experimental Environment

## E.1. Meta-World MT10

We evaluated LaGEA on the MetaWorld (Yu et al., 2020) MT-10 benchmark. Meta-World MT10 is a widely used benchmark for multi-task robotic manipulation, comprising ten goal-conditioned environments drawn from the broader Meta-World suite (Yu et al., 2020). All tasks are executed with a Sawyer robotic arm under a unified control interface: a $4D$ continuous action space (three Cartesian end-effector motions plus a gripper command) and a fixed $39D$ observation vector that encodes the end-effector, object, and goal states. Episodes are capped at 500 steps and share a common reward protocol across tasks, enabling a single policy to be trained and evaluated in a consistent manner.

Figure 6 depicts the ten tasks, and Table 10 lists the corresponding natural-language instructions that ground each goal succinctly. The suite spans fine motor skills (e.g., button pressing, peg insertion) as well as larger object interactions (e.g., reaching, opening/closing articulated objects), making MT10 a demanding testbed for generalization and multi-task policy learning.

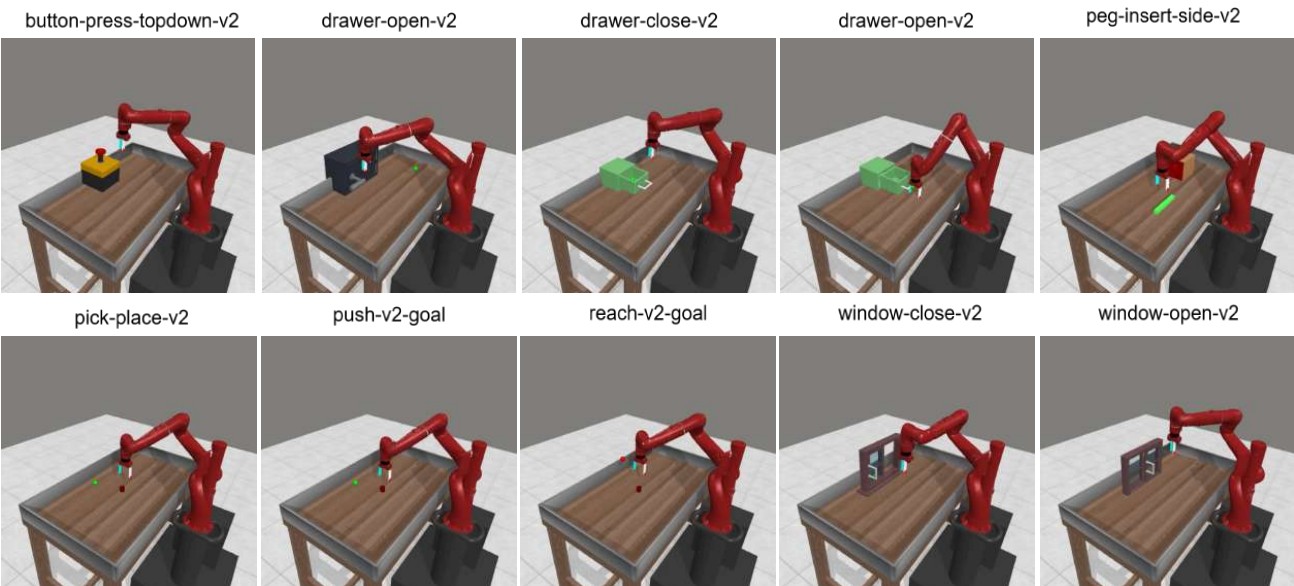

*Figure 6.* Meta-world MT10 benchmark tasks.

## E.2. Gymnasium-Robotics Fetch task

The Gymnasium-Robotics Fetch benchmark (Plappert et al., 2018) comprises four goal-conditioned tasks-Reach-v2, Push-v2, Slide-v2, and PickPlace-v2—executed with a simulated 7-DoF robotic arm equipped with a two-finger gripper. Actions are 4-dimensional Cartesian end-effector displacements (with gripper control where applicable), and observations follow the multi-goal API with {`observation`, `achieved_goal`, `desired_goal`}. Episodes are limited to 50 steps and use the standard sparse binary reward. Figure 7 illustrates the four environments, and Table 10 provides their corresponding natural-language task instructions.

| Environment | Text instruction |
|---|---|
| button-press-topdown-v2 | Press a button from the top. |
| door-open-v2 | Open a door with a revolving joint. |
| drawer-close-v2 | Push and close a drawer. |
| drawer-open-v2 | Open a drawer. |
| peg-insert-side-v2 | Insert the peg into the side hole. |
| pick-place-v2 | Pick up the puck and place it at the target. |
| push-v2 | Push the puck to the target position. |
| reach-v2 | Reach a goal position. |
| window-close-v2 | Push and close a window. |
| window-open-v2 | Push and open a window. |
| Fetch-Slide-v2 | Hit the puck so it slides and rests at the desired goal. |
| Fetch-Push-v2 | Push the box until it reaches the desired goal position. |
| Fetch-Reach-v2 | Move the gripper to the desired 3D goal position. |
| Fetch-PickPlace-v2 | Pick up the box and place it at the desired 3D goal position. |

*Table 10.* Environments and their text instructions of Meta-world MT10 and Gymnasium-Robotics Fetch benchmark tasks.

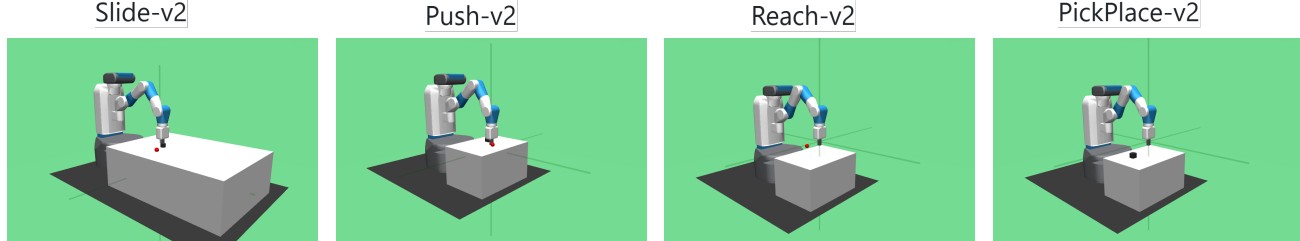

*Figure 7.* Gymnasium-Robotics Fetch benchmark tasks.

# F. Implementation Details

In our experiments, we use the latest Meta-World M10 (Yu et al., 2020) and Robotic Fetch (Plappert et al., 2018) environment. The main software versions are as follows:

- Python 3.11
- jax 0.4.16
- numpy 1.26.4
- flax 0.7.4

- gymnasium 0.29.1
- imageio 2.34.0
- mujoco 2.3.7
- optax 0.2.1

- torch 2.2.1
- torchvision 0.17.1
- jaxlib 0.4.16+cuda12.cudnn89
- gymnasium-robotics 1.2.4

# G. Algorithm

The pseudocode algorithm 1 formalizes the LAGEA training loop. Each episode, the policy collects a trajectory with RGB observations and a task instruction; we select a small set of key frames and query an instruction-tuned VLM (Qwen-2.5-VL-3B) (Bai et al., 2025) to produce a structured reflection (error code, key-frame indices, brief rationale). The instruction and reflection are encoded with a lightweight GPT-2 text encoder and paired with visual embeddings; a projection head is trained with a keyframe-gated alignment objective followed by a symmetric, weighted contrastive loss so that feedback becomes control-relevant. At training time we compute two potentials from these aligned embeddings: one that measures instruction–state goal agreement and one that measures transition consistency with the VLM diagnosis around the cited frames. We use only the change in these signals between successive states as a per-step shaping reward, add it to the environment reward with adaptive scaling and simple agreement gating (emphasizing failure episodes early and annealing

---

**Algorithm 1** LᴀGEA: Feedback–Grounded Reward Shaping (lean)

---

**Input** : Encoders $\Phi_I, \Phi_T, \Phi_F$; VLM $\mathcal{Q}$; goal image $o_g$; instruction $y$; replay buffer $\mathcal{D}$; episodes $N$

**Output** : trained policy $\pi$

1   **Initialize:** projection heads $E_i, E_t, E_f$; policy $\pi$; SAC learner.

    $z_g \leftarrow \text{norm}\big(E_i(\Phi_I(o_g))\big), \quad z_y \leftarrow \text{norm}\big(E_t(\Phi_T(y))\big).$

2   **for** $i = 1$ **to** $N$ **do**

3      /* Collect Trajectories Figure 1 */

      Roll out $\pi$ to obtain $\{(o_t, r_t^{\text{task}})\}_{t=0}^{T-1}$; push to $\mathcal{D}$.

      /*Key frames & per-step weights Section 3.1.2*/

      $x_t \leftarrow \Phi_I(o_t); \quad s_t \leftarrow \langle \text{norm}(E_i(x_t)), z_g \rangle;$

      $\mathcal{K} \leftarrow \text{GETKEYFRAMES}(s_{0:T-1}, M); \quad \widehat{w} \leftarrow \text{TRIANGULARWEIGHTS}(\mathcal{K}, h)$ (unit mean).

4      /*Structured episodic reflection Section 3.1.1*/

      Subsample $N$ frames; query $\mathcal{Q}$ with frames; encode feedback $z_f \leftarrow \text{norm}\big(E_f(\Phi_F(f))\big)$

5      /*Feedback alignment Section 3.1.3*/

      UPDATEFEEDBACKALIGNMENT$(E_i, E_f; \mathcal{D}, \hat{w})$;

      UPDATEFEEDBACKCONTRASTIVEWEIGHTED$(E_i, E_f; \mathcal{D}, \hat{w})$.

6      /*Dense Reward shaping Section 3.2*/

      **for** $t = 0$ **to** $T - 2$ **do**

7        $z_t \leftarrow \text{norm}(E_i(x_t)), \quad z_{t+1} \leftarrow \text{norm}(E_i(x_{t+1}));$

        Calculate goal delta; $r_t^{\text{goal}} \leftarrow \text{GOALDELTA}(z_t, z_{t+1}; z_y, z_g)$;

        Calculate feedback delta; $r_t^{\text{fb}} \leftarrow \text{FEEDBACKDELTA}(z_t, z_{t+1}; z_f)$;

        $\alpha \leftarrow \text{CLIP}\big(\alpha_{\text{base}} \cdot \frac{1 + \langle z_y, z_f \rangle}{2}, [\alpha_{\min}, \alpha_{\max}]\big)$;

        Calculate fused dense reward; $\tilde{r}_t \leftarrow (1-\alpha)\, r_t^{\text{goal}} + \alpha\, \widehat{w}_t\, r_t^{\text{fb}}$.

8      /*Adaptive reward shaping Section 3.2*/

      $\rho_t \leftarrow \text{ADAPTIVERHO}(\text{progress EMA / schedule})$;

      Overall reward; $r_t \leftarrow r_t^{\text{task}} + \rho_t\, \tilde{r}_t$.

9      /*Update SAC*/

      UPDATESAC$(\pi; \mathcal{D}, r_t)$.

---

over time), and update a standard SAC (Haarnoja et al., 2018) agent from a replay buffer with target networks.

## H. Feedback Pipeline

At the end of each episode, we run a deterministic key-frame selector over the image sequence to extract a compact set of causal moments $\mathcal{K}$. We then assemble a prompt with the task instruction, a compact error taxonomy, few-shot exemplars, and the selected frames, and query a frozen VLM (Qwen-2.5-VL-3B). The model is required to return a schema-constrained JSON with fields `outcome`, `primary_error{code, explanation}`, `secondary_factors`, `key_frame_indices`, `suggested_fix`, `confidence`, and `summary`. Responses are validated against the schema and retried on violations. Textual slots are normalized and embedded with a lightweight GPT-2 encoder to produce a feedback vector $f$ that is time-anchored via $\mathcal{K}$. This structured protocol reduces hallucination, yields feedback comparable across episodes and viewpoints, and makes the language signal embeddings directly consumable by the alignment and reward-shaping modules.

## I. Error Taxonomy

An error taxonomy is introduced to systematically characterize the types of failures observed in robot manipulation trajectories. This taxonomy provides discrete error codes that capture common failure modes in manipulation tasks, such as interacting with the wrong object, approaching from an incorrect direction, failing to establish a stable grasp, applying insufficient force, or drifting away from the intended goal. By mapping trajectories to these interpretable categories, we enable structured analysis of failure cases and facilitate targeted improvements in policy learning. Table 11 summarizes the error codes and their descriptions.

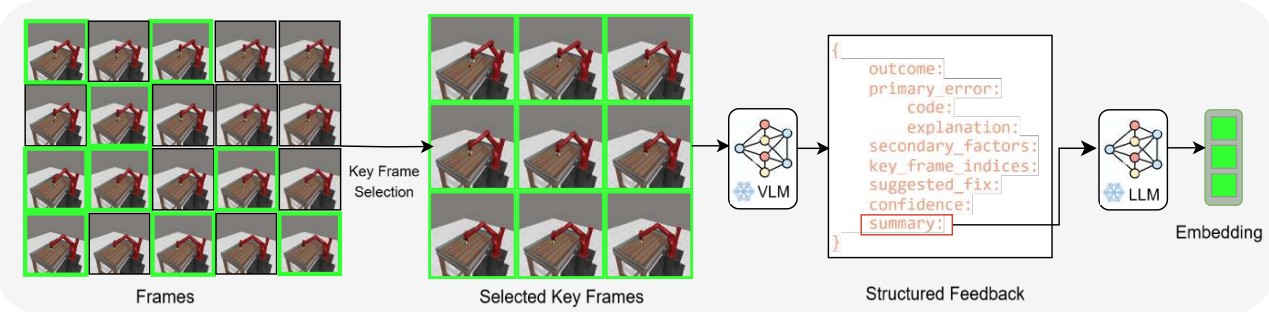

*Figure 8.* **Feedback Generation Pipeline**: Keyframes are selected from an episode, analyzed by a VLM to produce structured feedback text, which is then encoded into a final feedback embedding.

*Table 11.* Error codes and their descriptions.

| Error Code | Description |
|---|---|
| wrong_object | Interacted with the wrong object. |
| bad_approach_direction | Approached object from a wrong angle/direction. |
| failed_grasp | Contact without a stable grasp; slipped or never closed gripper appropriately. |
| insufficient_force | Touched correct object but did not exert proper motion/force. |
| drift_from_goal | Trajectories drifted away from the goal, no course correction. |

## J. Structured Feedback

Structured feedback mechanism constrains the VLM to produce precise, interpretable, and reproducible outputs. After each rollout, the model returns a JSON object that follows the schema shown in Figure 9, rather than free-form text. The schema records the task identifier, the binary outcome (success or failure), a single primary error code with a short explanation, optional secondary factors, key frames, a suggested fix, a confidence score, and a concise summary. This format anchors feedback to concrete evidence, keeps annotations consistent across episodes, and makes the signals directly usable for downstream analysis.

```
{
    task:  {string},
    outcome:  {success | failure},
    primary_error:  {
      code:  {error_code or success_code},
      explanation:  {one sentence explanation}
    },
    secondary_factors:  [{error_code, ...}],
    key_frame_indices:  [{int, int, int}],
    suggested_fix:  {string or (n/a)},
    confidence:  {float in [0,1]},
    summary:  {one sentence summary}
}
```

*Figure 9.* Schema for structured feedback returned by the VLM

Example structured feedback is shown for two Meta-World tasks - `button-press-topdown-v2` and `door-open-v2` - with two success cases in Figures 10 and Figure 11 and two failure cases in Figures 12 and Figure 13.

For the success cases, the schema assigns `primary_error.code=good_grasp`, with empty `secondary_factors`,

```
{
   task:  button-press-topdown-v2-goal-observable,
   outcome:  success,
   primary_error:  {
     code:  good_grasp,
     explanation:  The gripper successfully grasped the button.
   },
   secondary_factors:  [ ],
   key_frame_indices:  [12, 18],
   suggested_fix:  (n/a),
   confidence:  0.9,
   summary:  The agent succeeded because it grasped the button securely and
pressed it straight down, achieving the goal.
}
```

*Figure 10.* Success case with structured feedback for *button-press-topdown-v2-goal-observable* task.

```
{
   task:  door-open-v2-goal-observable,
   outcome:  success,
   primary_error:  {
     code:  good_grasp,
     explanation:  The gripper successfully grasped the black block and
opened its door.
   },
   secondary_factors:  [ ],
   key_frame_indices:  [9, 18, 27],
   suggested_fix:  (n/a),
   confidence:  0.9,
   summary:  The robot successfully opened the door of the black block by
grasping it and applying the appropriate force.
}
```

*Figure 11.* Success case with structured feedback for *door-open-v2-goal-observable* task.

high `confidence`, and `suggested_fix=(n/a)`. In `button-press-topdown-v2`, success is attributed to a secure grasp followed by a vertical, normal-aligned press that achieves the goal. In `door-open-v2`, success is similarly tied to a stable grasp on the handle and the application of sufficient force to open the door.

In the failure counterparts, the same schema yields concise, actionable diagnoses. For `button-press-topdown-v2`, `primary_error.code=bad_approach_direction` reflects a lateral approach that causes sliding; the prescribed fix is a topdown, normalaligned press. For `door-open-v2`, `primary_error.code=failed_grasp` with `insufficient_force` as a secondary factor attributes failure to unstable closure and inadequate actuation; the recommended remedy is a tighter grasp and sufficient force. Across both tasks, explanations remain succinct and suggested fixes translate diagnosis into concrete adjustments, ensuring comparability and evidential grounding within the structured format.

```
{
    task:  button-press-topdown-v2-goal-observable,
    outcome:  failure,
    primary_error:  {
      code:  bad_approach_direction,
      explanation:  The gripper came from the side, sliding off the button
instead of a vertical press.
    },
    secondary_factors:  [ ],
    key_frame_indices:  [18, 22],
    suggested_fix:  Approach from directly above the button; align gripper
normal to the button surface, then press straight down.,
    confidence:  0.85,
    summary:  The robot failed to press the button correctly because it
approached from the side instead of a vertical press.  This resulted in the
gripper sliding off the button.
}
```

*Figure 12.* Failure case with structured feedback for *button-press-topdown-v2-goal-observable* task.

```
{
    task:  door-open-v2-goal-observable,
    outcome:  failure,
    primary_error:  {
      code:  failed_grasp,
      explanation:  The gripper did not close properly around the door handle,
leading to a failed attempt to open the door.
    },
    secondary_factors:  [insufficient_force],
    key_frame_indices:  [16, 24],
    suggested_fix:  Ensure the gripper closes tightly around the door handle
and applies sufficient force.,
    confidence:  0.9,
    summary:  The agent failed to open the door as the gripper did not close
properly around the handle, indicating a failed grasp.
}
```

*Figure 13.* Failure case with structured feedback for *door-open-v2-goal-observable* task.

## K. Ablation

To quantify the contribution of each component in LΛGEA, we run controlled ablations with identical training settings, three random seeds per task, and we report mean (std.) success. All variants use the same encoders, SAC learner, and goal image; unless noted otherwise. The protocol followed for the ablation study is as follows:

**Feedback Alignment** Drop the multi-stage feedback−vision alignment and rely on frozen encoder similarities; tests whether learned alignment is required to obtain a control-relevant embedding geometry.

**Feedback Quality Ablation** Replace the schema-constrained (structured) feedback with unconstrained free-form VLM feedback text; measures the impact of feedback structure, reliability and hallucination on reward stability.

*Table 12.* Ablation results of LᴀGEA. Experiments were done using three different seeds. Results are averaged here.

| Task | Feedback Alignment | Feedback Quality Ablation | Keep all, drop adaptive $\rho$ | Drop all, keep adaptive $\rho$ | Key frame ablation | Delta reward ablation |
|---|---|---|---|---|---|---|
| button-press-topdown-v2-observable | 20 (34.64) | 10 (10) | 13.33(23.09) | 33.33(57.74) | 30 (51.96) | 30 (51.96) |
| drawer-open-v2-observable | 100 (0) | 96.67(5.77) | 100 (0) | 0 (0) | 76.67(40.41) | 100 (0) |
| door-open-v2-observable | 100 (0) | 100 (0) | 100 (0) | 0 (0) | 100 (0) | 76.67(40.41) |
| push-v2-hidden | 100 (0) | 66.67(57.74) | 66.67(57.74) | 33.33(57.74) | 100 (0) | 100 (0) |
| drawer-open-v2-hidden | 100 (0) | 100 (0) | 100 (0) | 33.33(57.74) | 100 (0) | 66.67(57.74) |
| door-open-v2-hidden | 100 (0) | 100 (0) | 100 (0) | 33.33(57.74) | 100 (0) | 100 (0) |

**Keep all, drop adaptive $\rho$** Use the full shaping signals but fix the mixing weight instead of scheduling it; probes the role of progress-aware scaling for stable learning.

**Drop all, keep adaptive $\rho$** Remove goal-/feedback-delta terms and keyframe gating while retaining the adaptive schedule (no auxiliary signal added); controls for the possibility that the schedule alone yields gains.

**Key frame ablation** Replace keyframe localization with uniform per-step weights; assesses the value of temporally focused credit assignment around causal moments.

**Delta reward ablation** Use absolute similarities instead of temporal deltas; tests whether potential-based differencing (which avoids static-state bias) is essential.

## L. Successful Trajectory Visualization

Figure 14 presents successful trajectory visualizations generated by LᴀGEA across nine environments from Meta-World MT10. Each trajectory illustrates how LᴀGEA effectively completes the corresponding manipulation task, highlighting its generalization ability across diverse settings. The only exception is `peg-insert-side-v2`, where LᴀGEA was unable to produce a successful episode; therefore, no trajectory is shown for this environment.

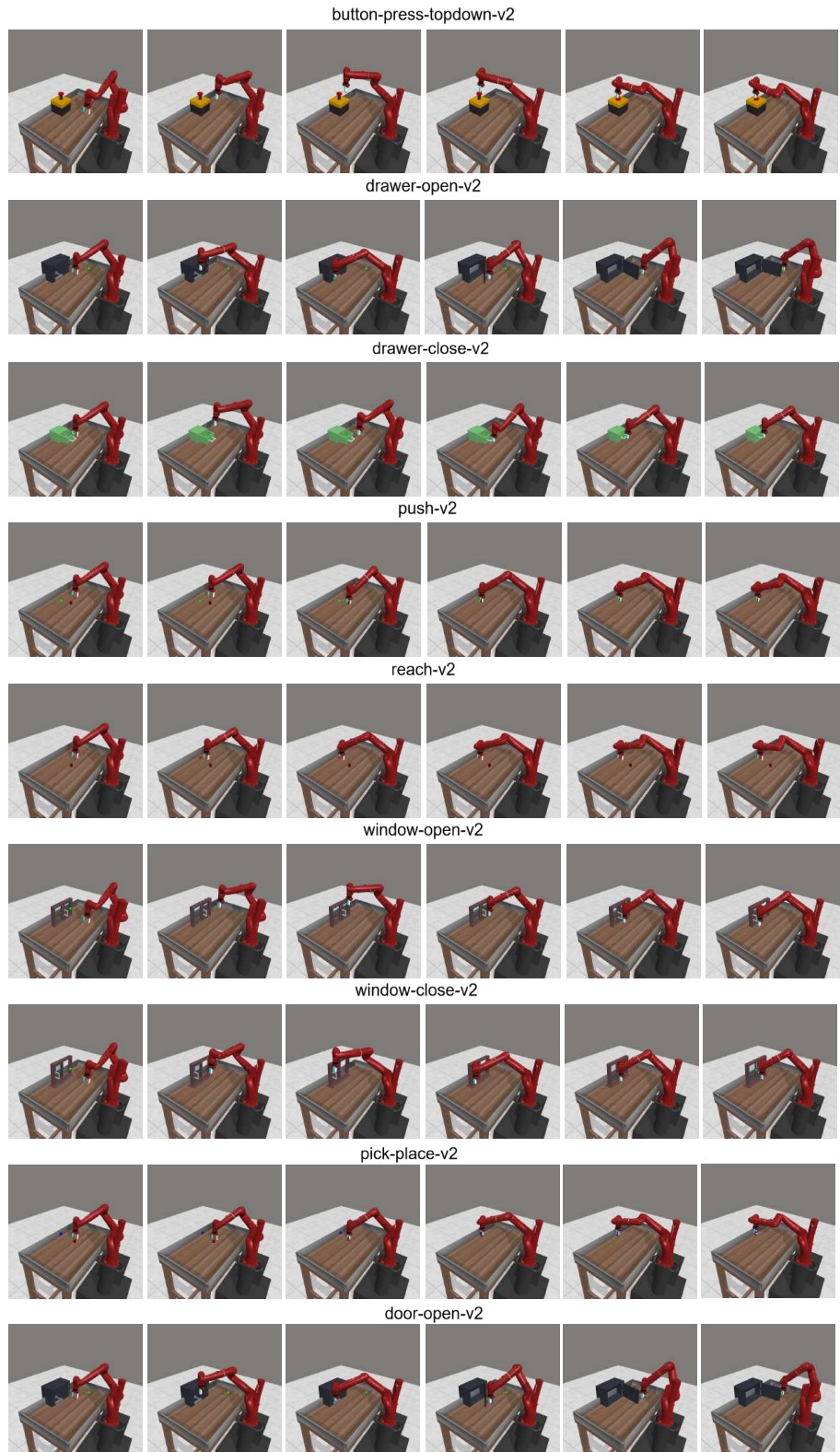

*Figure 14.* Visualization of successful trajectories using LᴀGEA on environments from Meta-World MT10 benchmark tasks.

