# OpenReview forum: "LAGEA: Language Guided Embodied Agents for Robotic Manipulation"
_ICML.cc/2026/Conference — ICML 2026 regular_

### Official Review · Reviewer_WudA · 2026-03-09

**Soundness:** 3
**Presentation:** 4
**Significance:** 3
**Originality:** 2
**Overall Recommendation:** 4
**Confidence:** 4

**Summary:**

LAGEA proposes a framework to enhance reinforcement learning (RL) for robotic manipulation by leveraging Vision-Language Models (VLMs) as high-level critics. When an agent fails a task, a VLM generates a structured natural language reflection that diagnoses the failure and localizes it temporally within the trajectory. These reflections are then converted into step-wise shaping rewards using a potential-based formulation, intended to provide denser signals for exploration. The method is evaluated on Meta-World MT10 and Robotic Fetch benchmarks, claiming improved success rates and faster convergence compared to existing VLM-reward baselines.Strengths

Soundness (Reward Formulation): The choice of potential-based reward shaping is technically sound. By using the change in potential from successive states rather than raw VLM scores, the authors mitigate the risk of "reward hacking" and stabilize the learning process against noisy VLM outputs.

**Compliance With Llm Reviewing Policy:**

Affirmed.

**Final Justification:**

The rebuttal demonstrates that LAGEA degrades gracefully under noisy feedback, which is practically valuable. However, it also reveals that precise VLM diagnosis contributes less than the paper implies, and the originality concern remains. We adjust our score from 3 (Weak Reject) to 4 (Weak Accept).

**Key Questions For Authors:**

1. Stochasticity of Feedback: How does the framework handle inconsistent feedback from the VLM?  If the VLM provides different rationales for identical failure trajectories, does the shifting reward potential destabilize the critic?

2. Ablation of Diagnosis Accuracy: Could the authors provide an experiment where the VLM feedback is replaced with "randomly chosen but task-relevant" reflections?  This would help isolate whether the gains come from the actual content of the diagnosis or simply from having a denser, language-aligned reward signal.

3. Real-world Latency: What is the average wall-clock time increase when using LAGEA compared to standard SAC/PPO?  Specifically, how does the VLM inference time scale as the number of failures increases during early exploration?

4. Handling Hallucinations: When a VLM "rationalizes" a failure incorrectly, does the adaptive coefficient effectively suppress the resulting noisy reward, or does the agent still attempt to optimize for the incorrect linguistic goal?

**Limitations:**

Yes. The authors acknowledge that smaller VLMs can hallucinate and that feedback reliability is a potential bottleneck.  However, more discussion on the energy/cost implications of using large model inference for RL reward generation would be beneficial.

**Strengths And Weaknesses:**

Strengths:
1. Soundness (Reward Formulation): The choice of potential-based reward shaping is technically sound. By using the change in potential from successive states rather than raw VLM scores, the authors mitigate the risk of "reward hacking" and stabilize the learning process against noisy VLM outputs.
2. Originality (Temporal Grounding): Unlike previous works that use VLMs for global episode scoring, LAGEA attempts to localize decisive moments in the trajectory. This granular approach to credit assignment is a significant step toward making language feedback actionable for low-level control.
3. Presentation: The motivation is well-articulated. The paper correctly identifies the "bottleneck" of VLM-derived rewards—namely, their susceptibility to hallucination and lack of step-wise precision—and builds a logical case for structured reflections.

Weaknesses:
1. Soundness (VLM Reliability): The system's performance is fundamentally tethered to the VLM’s ability to correctly diagnose failures. While "schema-constrained" reflections are used, there is a lack of rigorous analysis on how the policy reacts when the VLM provides a "false rationale" (e.g., misidentifying the cause of a grasp failure).  The robustness of the shaping signal under high VLM error rates remains an open question.
2. Significance (Inference Overhead): RL training often requires millions of steps. Calling a VLM for reflection and alignment after every failed episode introduces considerable computational latency. The paper does not provide a clear comparison of "wall-clock time" efficiency; a 9% success rate gain might not justify a 5x increase in training time.
3. Originality (Incremental Innovation): While the integration is neat, the individual components—VLM for feedback and reward shaping for RL—are well-established. The "Adaptive failure-aware coefficient" feels somewhat heuristic, and its theoretical necessity for convergence is not fully explored.

---

> ### Author Rebuttal · Authors · 2026-03-31
>
> We thank the reviewer **WudA** for the detailed and constructive feedback. We address each concern and question below.
>
> **1. VLM feedback stochasticity, diagnosis accuracy, and hallucination robustness**
> > "How does the framework handle ... potential destabilize the critic?"
>
> > "Could the authors provide an experiment ... language-aligned reward signal."
>
> > "The system's performance ... VLM error rates remains an open question."
>
> LAGEA is designed so that inconsistent VLM outputs do not directly become raw rewards. These three questions share a common empirical answer, so we address them together with two controlled experiments.
>
> i) Random task-relevant reflections. We replaced the VLM's feedback with randomly sampled reflections
>
> ii) Generic fallback. We replaced feedback with a generic fallback phrase ("the agent failed; adjust approach")
>
> All rebuttal experiments use three seeds. LAGEA (original) uses five seeds. We use the same five observation-based MT10 tasks already used in *Appendix C*.
>
> Task                               | Random Reflection  (STD) |Generic Reflection (STD)  | LaGEA  (STD)
> -----------------------------------|-----------------|------------------|--------------
> Button-press-topdown-v2-observable | 43.33 (20.55)   |  33.33 (17)      | 96 (8)
> Door-open-v2-observable            | 83.33 (4.71)    |  100 (0)         | 100 (0)
> Drawer-open-v2-observable          | 100 (0)         |  100 (0)         | 92 (9.8)
> Push-v2-observable                 | 3.33 (4.71)     |  0  (0)          | 12 (4)
> Window-open-v2-observable          | 100 (0)         |  100 (0)         | 100 (0)
> **Average**                        | **66**          |  **66.67**       | **80**
>
> Both corrupted baselines drop ~13–14% below LAGEA on average. Importantly, they do not catastrophically fail, which confirms two things simultaneously: (i) the VLM's actual diagnostic content contributes meaningfully to LAGEA's gains (as random/generic conditions show a clear gap), and (ii) the framework is not brittle to occasional hallucinations, because the structured schema and adaptive $\rho$ together provide a safety net.
>
> ---
>
> **2. Wall-clock time and inference overhead**
> > "What is the average wall-clock time ... increases during early exploration?"
>
> > "RL training often requires ... might not justify a 5x increase in training time."
>
> Despite VLM inference accounting for 43.5% of wall-clock time, LaGEA converges faster on average than FuRL (92.36 vs 94.90 minutes, `Appendix C.2 Table 7)`, because it requires far fewer environment steps to solve each task. VLM inference overhead is real; however, **LaGEA's** improved sample efficiency compensates for the reflection overhead.
>
> ---
>
> **3. Hallucination suppression and theoretical necessity of adaptive $\rho$**
> > "When a VLM "rationalizes" a failure ... optimize for the incorrect linguistic goal?"
>
> > "While the integration is neat ... theoretical necessity for convergence is not fully explored."
>
> Our claim is modest: adaptive $\rho$ is a stabilization mechanism, not a theorem-derived guarantee. It only applies shaping on failures and anneals the shaping strength as competence improves, so the critic is progressively less exposed to noisy language reward once the policy begins to solve the task. This is exactly how the method is defined in `Sec. 3.3`.
>
> Empirically, the paper shows that removing adaptive shaping hurts performance in the reward ablation `(Fig. 4b).` Under maximally inconsistent or generic feedback (experiments on **answer 1**), LaGEA degrades gracefully to 66–67% rather than catastrophically, which is exactly the behavior expected if the adaptive $\rho$ and the alignment module are effectively suppressing semantically incoherent signals.
>
> *On the theoretical necessity of ρ:* the ablation "Drop adaptive $\rho$" in `Table 10` and `Figure 4b` shows that fixing $\rho$ to a constant causes a measurable performance drop, particularly on tasks requiring extended exploration. Our intuition and motivation behind adaptive $\rho$ is that a constant shaping weight cannot both provide strong guidance early and avoid overpowering the task reward late.
>
> ---
>
> We hope these experiments and clarifications address the reviewer's core concerns. We would greatly appreciate an updated score if the reviewer finds the evidence satisfactory.

---

> > ### Author Rebuttal · Reviewer_WudA · 2026-04-04
> >
> > We thank the authors for the supplementary experiments.  The rebuttal demonstrates that LAGEA degrades gracefully under noisy feedback, which is practically valuable. However, it also reveals that precise VLM diagnosis contributes less than the paper implies, and the originality concern remains. We adjust our score from 3 (Weak Reject) to 4 (Weak Accept).

---

### Official Review · Reviewer_1sb4 · 2026-03-12

**Soundness:** 4
**Presentation:** 2
**Significance:** 4
**Originality:** 4
**Overall Recommendation:** 5
**Confidence:** 3

**Summary:**

The paper proposes LAGEA, aiming to address the issue that agents struggle to learn from their own errors in a principled manner during manipulation tasks. LAGEA first uses a Vision-Language Model to summarize the episode and generate structured feedback, and computes the step-wise saliency of the frames in the trajectory to extract keyframes. After aligning the text feedback with the visual states in the embedding space, it finally calculates a step-wise shaping reward based on goal progress and feedback consistency. An adaptive scheduling mechanism dynamically allocates reward guidance based on the agent's learning progress. Experiments on METAWORLD MT10 and FETCH TASKS demonstrate the effectiveness of LAGEA.

**Compliance With Llm Reviewing Policy:**

Affirmed.

**Final Justification:**

I am glad to see that the author can correct these issues. The author's response can address some of my concerns, so I raise my score to 5.

**Key Questions For Authors:**

Optimize Figure 1 to make the overall model framework easier to understand. Meanwhile, improve the writing coherence among the various modules in the methodology section.

Attempt to add some real-world experiments to demonstrate the effectiveness of LAGEA.

Although the code release was later than the ICML submission deadline, could you consider including a comparison with Robo-Dopamine [1] in the experiments? This is not a mandatory request, but I believe the supplementary comparative experiments would further highlight the value of this paper.

Refs:

[1] Tan, Huajie and Chen, Sixiang, Robo-Dopamine: General Process Reward Modeling for High-Precision Robotic Manipulation, arXiv preprint arXiv:2512.23703, 2025

**Limitations:**

yes

**Strengths And Weaknesses:**

Strengths:

LAGEA innovatively proposes Structured Feedback Generation, mapping complex visual failures to specific discrete error codes and fixed formats. While ensuring the consistency of the feedback, it provides clear, causal manipulation guidance, endowing the model with the ability to learn and recover from errors.

The compact error taxonomy strictly constrains the VLM output through a fixed JSON format and selectable options, avoiding the drift and hallucination of free text to improve the stability of the VLM reward signal.

Compared to previous "goal proximity reward" calculation methods, LAGEA further expands reward generation, realizing dynamic reward adjustment through temporal-difference-based incremental goal rewards and incremental feedback rewards. The introduction of the adaptive adjustment coefficient $\rho_t$ also enables the VLM's reward information to be applied more reasonably.

LAGEA innovatively and tightly integrates two major problems in robot manipulation: "dense rewards" and "recovering from failures," providing an important paradigm for future research in related fields, which is of great significance.

Weaknesses:
As the paper involves many formulas, some of the expressions are not very clear, which hinders the reading process (e.g., the expression of $u_t$ in Section 3.1.3). Furthermore, the writing logic between the various modules in the methodology section is not coherent enough, and the pipeline shown in Figure 1 is not easy to understand.

Although mentioned by the authors in the limitations, the lack of real-world scene experiments still heavily impacts the value of the model. In addition, LAGEA's experiments rely on a single-view camera to extract keyframes and evaluate states. Occlusion problems in simulation environments are usually relatively simple, but the paper does not design specific experiments to verify whether the single-view VLM can still provide accurate feedback when the robotic arm experiences severe self-occlusion or environmental occlusion.

---

> ### Author Rebuttal · Authors · 2026-03-30
>
> We thank reviewer **1sb4** for the excellent scores and the constructive feedback. We address each concern below.
>
> **1. Presentation/clarity of Fig. 1 and Sec. 3.**
>
> > "As the paper involves many formulas ... Figure 1 is not easy to understand."
>
> > "Optimize Figure 1 to make ... various modules in the methodology section."
>
>
> We agree that the presentation can be improved. The intended logic of the method is the following: (i) generate a structured episodic reflection, (ii) localize where that feedback should matter through keyframe weights, (iii) align feedback and visual states in a shared space, and (iv) convert goal progress and feedback consistency into delta-based shaping rewards with adaptive scaling. This flow is already present in Fig. 1, Fig. 2, and Algorithm 1. In the revised manuscript, we will bring simple changes to improve readability.
>
> For the notation issue: in Sec. 3.1.3, `$u_t$` denotes the **per-step alignment weight**, i.e. *keyframe saliency multiplied by goal proximity* and then renormalized to unit mean, so that alignment gradients concentrate on causal, near-goal moments. The intent is that only timesteps that are both near a keyframe and near the goal drive the alignment gradients. The current text states this too compactly; we will rewrite it explicitly and simplify the notation.
>
> ---
>
> **2. Why not compare with Robo-Dopamine?**
>
> > "Although the code release was later than ... further highlight the value of this paper."
>
> We appreciate the suggestion and agree **Robo-Dopamine** is highly relevant. However, it studies a materially different setting from ours. Robo-Dopamine is built around a general process reward model trained on a *3,400+* hour dataset, uses multi-view inputs, performs one-shot task adaptation from a single expert trajectory, and is paired with a different Dopamine-RL framework for high-precision manipulation.
>
> `LAGEA`, in contrast, studies online sparse-reward robotic manipulation using single-view episodic reflections and no large pre-trained process reward model or any task-specific annotation. The two methods address complementary challenges (annotation-efficient dense rewards vs. zero-annotation episodic reflection). A direct comparison would therefore conflate two very different sources of supervision and prior data, rather than isolate the contribution of our method. For this reason, we chose baselines in the similar online manipulation regime. We will clarify this relation explicitly in the revised related work.
>
> **3. Single-view setting/occlusion/viewpoint robustness.**
>
> > "Although mentioned by the authors ... severe self-occlusion or environmental occlusion."
>
> We agree this is an important concern. While we do not claim to solve severe self-occlusion or cluttered-scene occlusion in this work, we added a **viewpoint-shift** experiment to test robustness to camera changes.
>
> `LaGEA` is trained entirely at Viewpoint 2 (Top-Left diagonal-front camera) and evaluated zero-shot at three held-out viewpoints (0, 1, and 3).
>
> Viewpoint 0 = Directly overhead, looking straight down, Viewpoint 1 = Front-left diagonal, slightly elevated; Viewpoint 3 = Behind-left diagonal, elevated top-view.
>
> These four viewpoints represent meaningfully different spatial perspectives. For this study, we use the same five observation-based MT10 tasks already used in *Appendix C*. We chose this subset because it is already the paper’s observation-based robustness subset, and because it covers a compact but varied range of manipulation types.
>
> Viewpoint {0,1,3} experiments were done using three seeds. The original paper LaGEA (Viewpoint 2) experiment was done using five seeds.
>
> | Task                                  | Viewpoint 0 (STD) | Viewpoint 1 (STD) | Viewpoint 3 (STD) |  LaGEA  (STD)   |
> |---------------------------------------|-------------------|-------------------|-------------------|-----------------|
> | Button-press-topdown-v2-observable    | 83.33 (12.47)     | 80 (16.33)        | 83.33 (17)        | 96 (8)          |
> | Door-open-v2-observable               | 100 (0)           | 100 (0)           | 100 (0)           | 100 (0)         |
> | Drawer-open-v2-observable             | 100 (0)           | 100 (0)           | 100 (0)           | 92 (9.8)        |
> | Push-v2-observable                    | 16.67 (4.71)      | 10 (0)            | 13.33 (4.71)      | 12 (4)          |
> | Window-open-v2-observable             | 96.67 (4.71)      | 96.67 (4.71)      | 100 (0)           | 100 (0)         |
> | **Average**                           | **79.33**         | **77.33**         | **79.33**         | **80**          |
>
> **LaGEA** sustains strong performance across all three unseen viewpoints (77–79% average), with negligible variance between them, although severe occlusion remains an open limitation.
>
> We hope these clarifications and the new viewpoint-shift results address the reviewer’s concerns. If the reviewer finds them satisfactory, we would greatly appreciate an updated score.

---

> > ### Author Rebuttal · Reviewer_1sb4 · 2026-04-03
> >
> > I am glad to see that the author can correct these issues. The author's response can address some of my concerns, so I raise my score to 5.

---

### Official Review · Reviewer_YHzz · 2026-03-13

**Soundness:** 3
**Presentation:** 4
**Significance:** 3
**Originality:** 3
**Overall Recommendation:** 5
**Confidence:** 4

**Summary:**

This paper introduces LAGEA, which uses VLM-generated reflections as reward shaping signals for reinforcement learning in robotic manipulation. The framework selects causally significant frames, generates feedback via a language model on these frames, aligns feedback embeddings with visual reps, and shapes rewards this new source of dense feedback. Experiments on Meta-World and Robotic Fetch show improvements over a range of baselines.

**Compliance With Llm Reviewing Policy:**

Affirmed.

**Final Justification:**

My concerns are sufficiently addressed. I maintain my score.

**Key Questions For Authors:**

1. Could comparisons against more related work, such as those in Appendix B under "VLMs for RL" be included as baselines? Or if not applicable, could an explanation be provided for why?

2. Related to question 1 above, are there any prior work that achieve success on the hardest tasks such as peg-insert-side-v2 and pick-place-v2? Why or why not?

3. Could we see additional testing on envs with partial observability or some kind of domain shift requiring generalization? This would stress test the capabilities of the VLM component of this method.

**Limitations:**

yes

**Strengths And Weaknesses:**

Strengths:

- The insight that feedback should be causal and temporally specific is sound and addresses an important weakness in prior VLM-reward methods.
- In addition, we see in Table 4 that structured feedback clearly outperforms freeform feedback (roughly 98% to 78%)
- Section 4.3 performs very detailed ablations of all the method's major components. Keyframe ablation seems particularly important.
- Results in tables 5 and 6 further support that the results presented in this paper are not isolated to the specific models tested. Instead, the paper presents a framework whose gains are generally agnostic to model choice.

Weaknesses:

- The paper compares against SAC, LIV, LIV-Proj, Relay, and FuRL. However, several recent methods cited in related work such as LIFT and RL-VLM-F are not compared against.
- All tasks use Mujoco in scenes with limited visual complexity/variation for the VLM to handle. Unclear how this method holds up in situations where there is visual ambiguity, occlusion, or varied camera viewpoints.

---

> ### Author Rebuttal · Authors · 2026-03-30
>
> We thank Reviewer **YHzz** for the positive assessment and for highlighting the key strengths of the paper. We address each concern and question below:
>
> **1. Why not compare directly against RL-VLM-F/LiFT?**
> > "The paper compares ... LIFT and RL-VLM-F are not compared against."
>
> > "Could comparisons ... explanation be provided for why?"
>
> We agree these are relevant related works. However, we do not view them as directly comparable baselines for the question studied in this paper.
>
> *RL-VLM-F* is a preference-based RL method which queries a VLM to compare pairs of images and learns a reward model from those pairwise labels via preference-based RL. RL-VLM-F's training regime is fundamentally different from `LaGEA's` framework, which generates episodic reflection and operates without pairwise preference collection. Comparing the two under sparse-reward protocol would require a substantial reimplementation of RL-VLM-F; making a plug-in comparison non-trivial. LaGEA subsumes the spirit of RL-VLM-F's goal (VLM-driven reward without human labels) while operating online, episodically, without a separate reward learning phase.
>
> *LIFT* targets unsupervised skill discovery in open-ended environments. It uses a VLM to propose task instructions and reward their completion in an environment without a fixed task set. Our setting is the opposite: a fixed, sparse-reward manipulation benchmark with predefined tasks. LIFT's contribution is not reward shaping for known tasks but instruction generation for unknown ones. It does not report results on Meta-World MT10 and cannot be meaningfully compared under our protocol. Our chosen baselines operate under the identical sparse-reward protocol and represent the direct line of work (VLM-RL reward shaping) that *LaGEA* advances.
>
> ---
>
> **2. Are there prior methods that succeed on peg-insert-side-v2 / pick-place-v2? Why are these still hard?**
> > "Related to question 1 above ... pick-place-v2? Why or why not?"
>
> To the best of our knowledge, no prior VLM-reward method has reported strong results on these exact MT10 hardest tasks under the same online sparse-reward setting.
>
> Our interpretation is that these environments expose the main current limitation of language-guided shaping: LAGEA helps with high-level correction and sparse-reward exploration, but the final success still depends on fine grasp/contact geometry and narrow alignment tolerances, which are hard to resolve from a single RGB view and coarse semantic feedback alone. On top of this, these tasks require fine-grained precision-sensitive control (reach - grasp - transport) that is inherently difficult to solve from sparse rewards and pixel observations alone.
>
> ---
>
> **3. Additional robustness under viewpoint shift/domain shift.**
> > "All tasks use Mujoco in scenes ... varied camera viewpoints."
>
> > "Could we see additional testing on envs ... VLM component of this method"
>
> We agree that this is an important stress test, and we added a **viewpoint-shift** experiment. `LaGEA` is trained entirely at Viewpoint 2 (Top-Left diagonal-front camera) and evaluated zero-shot at three held-out viewpoints (0, 1, and 3).
>
> Viewpoint 0 = Directly overhead, looking straight down, Viewpoint 1 = Front-left diagonal, slightly elevated; Viewpoint 3 = Behind-left diagonal, elevated top-view.
>
> These four viewpoints represent meaningfully different spatial perspectives. For this study, we use the same five observation-based MT10 tasks already used in *Appendix C*. We chose this subset because it is already the paper’s observation-based robustness subset, and because it covers a compact but varied range of manipulation types.
>
> Viewpoint {0,1,3} experiments were done using three seeds. The original paper LaGEA (Viewpoint 2) experiment was done using five seeds.
>
> | Task                                  | Viewpoint 0 (STD) | Viewpoint 1 (STD) | Viewpoint 3 (STD) |  LaGEA  (STD)   |
> |---------------------------------------|-------------------|-------------------|-------------------|-----------------|
> | Button-press-topdown-v2-observable    | 83.33 (12.47)     | 80 (16.33)        | 83.33 (17)        | 96 (8)          |
> | Door-open-v2-observable               | 100 (0)           | 100 (0)           | 100 (0)           | 100 (0)         |
> | Drawer-open-v2-observable             | 100 (0)           | 100 (0)           | 100 (0)           | 92 (9.8)        |
> | Push-v2-observable                    | 16.67 (4.71)      | 10 (0)            | 13.33 (4.71)      | 12 (4)          |
> | Window-open-v2-observable             | 96.67 (4.71)      | 96.67 (4.71)      | 100 (0)           | 100 (0)         |
> | **Average**                           | **79.33**         | **77.33**         | **79.33**         | **80**          |
>
> LaGEA sustains strong performance across all three unseen viewpoints (77–79% average), with negligible variance between them.
>
> We hope these clarifications and the new viewpoint-shift results address the reviewer’s concerns.

---

> > ### Author Rebuttal · Reviewer_YHzz · 2026-04-03
> >
> > Thanks to the authors for their response. I maintain my score.

---

### Official Review · Reviewer_qAX8 · 2026-03-13

**Soundness:** 2
**Presentation:** 2
**Significance:** 2
**Originality:** 3
**Overall Recommendation:** 3
**Confidence:** 4

**Summary:**

Instead of using language only to specify goals, the paper uses language as post-episode error feedback that is temporally grounded and converted into dense RL shaping signals. It proposes LAGEA, an embodied VLM-RL framework that generates causal episodic feedback which
are localized in time to turn failures into guidance and
improve recovery after near misses. On the Meta-World MT10 and
Robotic Fetch benchmark, LAGEA improves average success over SOTA by 9.0%
on random goals, 5.3% on fixed goals, and 17%
on fetch tasks with faster convergence.

**Compliance With Llm Reviewing Policy:**

Affirmed.

**Final Justification:**

Part of my questions are resolved. However, I am keeping my score unchanged because the evaluation scope is still limited. The experiments do not establish robustness to domain shift, longer-horizon settings, or real robots.

**Key Questions For Authors:**

1. How much of the gain is really from language content? Would similar gains arise from a non-language structured error classifier, or from improved shaping heuristics alone?

2. Can the authors compare the selected keyframes against human annotations, oracle key moments, random keyframes, or attention-based alternatives?

3. How portable is the error taxonomy?

4. How scalable is the method computationally, especially on long-horizon tasks, higher-frame-rate rollouts, or larger training runs where per-episode VLM reflection becomes a bottleneck?

**Limitations:**

The authors did not discuss the limitations and potential negative societal impact of their work. The limitations are the limited external validity, compute cost, and insufficient proof that the key language/causal components are doing what the paper claims.

**Strengths And Weaknesses:**

## Strengths

1. The paper's target problem is clear and well motivated.

2. The pipeline to turn episodic reflections into temporally grounded step-wise rewards is quite novel and its design intuition is strong.

3.  On the Meta-World MT10 and Robotic Fetch benchmark, LAGEA improves average success over SOTA by 9.0% on random goals, 5.3% on fixed goals, and 17% on fetch tasks with faster convergence. The ablation studies also support the claim.

## Weaknesses

1. The evaluation scope is limited. The experiments do not establish robustness to domain shift, longer-horizon settings, or real robots.

2. The method requires per-episode VLM querying on sampled frames, plus embedding and alignment machinery. This could strongly increase the wall-clock time for longer-horizon or higher-throughput training.

3. The paper argues keyframes identify decisive moments, but the support is mainly heuristic plus ablation. There is no oracle comparison, human annotation, or causal validation.

4. The system is fairly complex and somewhat engineering-heavy.

5. The model choice is narrow, i.e.,  Qwen2.5-VL-3B + GPT-2 combination. Larger and more recent models should be evaluated.

6. The method still fails on hard precision-sensitive tasks like peg-insert-side-v2. This might suggest the approach helps with high-level correction but may not solve fine-grained control bottlenecks.

---

> ### Author Rebuttal · Authors · 2026-03-31
>
> We thank reviewer **qAX8** for the detailed evaluation. We address each concern directly below.
>
> **1. How much gain is from language content? Would a symbolic classifier suffice?**
> > "How much of the gain ... shaping heuristics alone?"
>
> We ran a targeted experiment to answer it directly. To isolate language more directly, we added a symbolic-code baseline that replaces the feedback summary with only the primary error code, encoded as a learned one-hot embedding.
>
> Code-only experiments were done using three seeds. The original paper LaGEA experiments were done using five seeds.
>
> Task     | Code-Only (STD)| LaGEA (STD)
> -------------|----------------|--------------
> Button-press-topdown-v2-observable | 33.33 (12.47)  | 96 (8)
> Door-open-v2-observable   | 70 (8.16)      | 100 (0)
> Drawer-open-v2-observable    | 100 (0)        | 92 (9.8)
> Push-v2-observable    | 3.33 (4.71)    | 12 (4)
> Window-open-v2-observable    | 90 (8.16)      | 100 (0)
> **Average**     | **59.33**   | **80**
>
> On the same 5 task subset used in Appendix C, this yields 59.33 average, versus 80 for LAGEA. Coarse symbolic failure label leaves a large gap to the full natural-language reflection. Hence, the gain is not from shaping heuristics alone; the richer language signal carries useful information beyond a discrete code.
>
> ---
>
> **2. Can the selected keyframes be compared to random/uniform alternatives?**
> > "Can the authors compare ... attention-based alternatives?"
>
> > "The paper argues keyframes ...  or causal validation."
>
> Original paper shows - `sec. 4.3.2` - that removing keyframing causes catastrophic failure on `Drawer-Open`, indicating that temporally focused credit assignment is necessary. On top of this, we ran a direct keyframe comparison across randomly sampled frames, uniformly sampled frames, and LaGEA keyframes on the same five tasks as Appendix C. Again, the random seeds for the rebuttal experiments are three.
>
> Task       | Random KF (STD) |Uniform KF (STD)  | LaGEA  (STD)
> -----------------------------------|-----------------|------------------|--------------
> Button-press-topdown-v2-observable | 36.67 (17)      |  50 (24.49)      | 96 (8)
> Door-open-v2-observable            | 93.33 (9.43)    |  100 (0)         | 100 (0)
> Drawer-open-v2-observable          | 100 (0)         |  80 (16.33)      | 92 (9.8)
> Push-v2-observable     | 10 (0)          |  6.67 (9.43)     | 12 (4)
> Window-open-v2-observable          | 100 (0)         |  100 (0)         | 100 (0)
> **Average**    | **68**          |  **67.33**       | **80**
>
> **LAGEA** outperforms both random and uniformly-spaced keyframes on average. Our selector is stronger because it concentrates credit around causal moments derived from the goal-similarity trajectory, rather than spreading signal uniformly or stochastically.
>
> ---
>
> **3. How portable is the error taxonomy?**
>
> The taxonomy was designed around general manipulation primitives that recur across virtually all manipulation tasks. It is reused across all MT10/Fetch tasks under the same fixed schema.
>
> We therefore view it as portable within robotic manipulation. At the same time, we do not claim it is complete for all real-world manipulation; for broader domains, it would need to be extended, which is straightforward because the schema is modular.
>
>
> **4. Scalability and computational cost.**
> > "How scalable is the ... becomes a bottleneck?"
>
> > "The method requires per-episode ... higher-throughput training."
>
> Despite VLM inference accounting for 43.5% of wall-clock time, LaGEA converges faster on average than FuRL (92.36 vs 94.90 minutes, `Appendix C.2 Table 7`), because it requires far fewer environment steps to solve each task. VLM inference overhead is real; however, LaGEA's improved sample efficiency compensates for the reflection overhead.
>
> ---
>
> **5. System complexity and model choice.**
> > "The model choice is narrow ..."
>
> **Appendix C (Tables 5–6)** reports results across four recent VLM backbones and four recent text encoders. Every combination outperforms SAC and FuRL, and the full LAGEA framework achieves the best average across all combinations. We hope this addresses the `narrow model choice` concern.
>
> **6: Failure on peg-insert-side-v2.**
> > "The method still fails on ... fine-grained control bottlenecks."
>
> We agree this is a current limitation. The peg-insert task requires sub-centimeter alignment tolerances under partial self-occlusion. Gripper blocks the view of the peg-hole contact point from the training viewpoint. As a result, the VLM consistently misclassifies the failure as the hole is not visible at the decisive moment.
>
> Failure reasoning on fine-grained control tasks is also discussed in the reviewer **YHzz**'s rebuttal point **`#2`**.
>
> ---
>
> We hope these new experiments and clarifications address the reviewer's core concerns, and we would greatly appreciate an updated score if the reviewer finds them satisfactory.

---

> > ### Author Rebuttal · Reviewer_qAX8 · 2026-04-04
> >
> > Thanks to the authors for the reply. My concerns are partly resolved, and the rebuttal makes the paper stronger. Part of my questions are resolved. However, I am keeping my score unchanged because the evaluation scope is still limited. The experiments do not establish robustness to domain shift, longer-horizon settings, or real robots.

---

### Decision · Program_Chairs · 2026-04-30

**Decision:**

Accept (regular)

**Comment:**

Reviewers agreed that the authors propose a novel pipeline to turn episodic reflections into temporally grounded step-wise rewards. The paper's target problem is clear and well motivated. Most of the reviewers' comments were addressed during the discussion with the authors. The main issues remaining after the discussion phase relate to the limitations of the experimental results. That chill notes that the experiments do not establish robustness to domain shift, longer-horizon settings, or real robots. However, the authors' main contribution is still confirmed by current experiments, and the work does not relate to robotics; it is not entirely correct to require experiments on a real robot. I believe the work deserves acceptance at the conference.